# C/EBPδ-induced epigenetic changes control the dynamic gene transcription of *S100a8* and *S100a9*

Saskia-Larissa Jauch-Speer[1], Marisol Herrera-Rivero[2], Nadine Ludwig[3], Bruna Caroline Véras De Carvalho[1], Leonie Martens[1,2], Jonas Wolf[1], Achmet Imam Chasan[1], Anika Witten[2,4], Birgit Markus[5], Bernhard Schieffer[5], Thomas Vogl[1], Jan Rossaint[3], Monika Stoll[2,6], Johannes Roth[1]*, Olesja Fehler[1]

[1]Institute of Immunology, University of Münster, Münster, Germany; [2]Department of Genetic Epidemiology, Institute of Human Genetics, University of Münster, Münster, Germany; [3]Department of Anesthesiology, Intensive Care and Pain Medicine, University Hospital Münster, Münster, Germany; [4]Core Facility Genomics, Medical Faculty Münster, University of Münster, Münster, Germany; [5]Clinic for Cardiology, Angiology and Internal Intensive Medicine, University Hospital Marburg, Marburg, Germany; [6]CARIM Cardiovascular Research School, Department of Biochemistry, Genetic Epidemiology and Statistical Genetics, Maastricht University, Maastricht, Netherlands

*For correspondence:
rothj@uni-muenster.de

**Abstract** The proinflammatory alarmins S100A8 and S100A9 are among the most abundant proteins in neutrophils and monocytes but are completely silenced after differentiation to macrophages. The molecular mechanisms of the extraordinarily dynamic transcriptional regulation of *S100a8* and *S100a9* genes, however, are only barely understood. Using an unbiased genome-wide CRISPR/Cas9 knockout (KO)-based screening approach in immortalized murine monocytes, we identified the transcription factor C/EBPδ as a central regulator of *S100a8* and *S100a9* expression. We showed that S100A8/A9 expression and thereby neutrophil recruitment and cytokine release were decreased in C/EBPδ KO mice in a mouse model of acute lung inflammation. *S100a8* and *S100a9* expression was further controlled by the C/EBPδ antagonists ATF3 and FBXW7. We confirmed the clinical relevance of this regulatory network in subpopulations of human monocytes in a clinical cohort of cardiovascular patients. Moreover, we identified specific C/EBPδ-binding sites within *S100a8* and *S100a9* promoter regions, and demonstrated that C/EBPδ-dependent JMJD3-mediated demethylation of H3K27me$_3$ is indispensable for their expression. Overall, our work uncovered C/EBPδ as a novel regulator of *S100a8* and *S100a9* expression. Therefore, C/EBPδ represents a promising target for modulation of inflammatory conditions that are characterized by *S100a8* and *S100a9* overexpression.

## Editor's evaluation

The study uses an elegant CRISPR/Cas9 screening approach in a myeloid cell line to identify the underlying regulators of the alarmins S100A8 and S100A9, which amplify inflammation. This approach identified the transcription factor C/EBP-δ as a regulator of S100A8 and S100A9 expression in the myeloid cell line and also showed a correlation between the expression levels of C/EBP-δ and the alarmins in patient samples of peripheral blood mononuclear cells. Furthermore, the authors also validate their findings in primary monocytes using mice with genetic C/EBP-δ deletion. This work will be of significant interest to researchers studying the regulation of immune responses and

inflammation, and it also highlights how unbiased CRISPR/Cas9 screening can lead to novel mechanistic insights in myeloid cells.

## Introduction

As the first line of immune defence, both monocytes and neutrophils are important for the modulation of the innate immune response. To amplify the immune response at sites of inflammation, the activation of further immune cells is required, mediated by the release of signaling molecules such as chemokines and DAMPs (damage-associated molecular patterns). The two members of the S100 family, S100A8 and S100A9, also termed myeloid-related proteins 8 and 14 (MRP8 and MRP14), respectively, belong to the group of DAMPs or so-called alarmins. Their primary expression is referred to myeloid lineage-derived cells, particularly neutrophils and monocytes, where S100A8 and S100A9 are predominantly present as a heterodimeric complex, also called calprotectin (*Austermann et al., 2018*).

Intracellularly, S100A8/A9 complexes represent up to 40% of the soluble protein content in neutrophils and about 5% in monocytes (*Hessian et al., 1993*). However, in mature macrophages, protein and mRNA expression of these factors is completely downregulated. This data indicates that expression of *S100A8* and *S100A9* is controlled by the most dynamic promoters in myeloid cells. The S100A8/A9 complex interacts with the cytoskeleton in a calcium-dependent manner. Calcium-induced (S100A8/A9)$_2$ tetramer promotes tubulin polymerization and microtubule bundling, thereby affecting transendothelial migration of phagocytes (*Leukert et al., 2006*). During inflammation or tissue damage, S100A8/A9 is actively secreted by neutrophils and monocytes, and represents the most abundant DAMP/alarmin activating inflammatory processes in infection, cancer, autoimmunity, and cardiovascular diseases. The S100A8/A9 complex is recognized by Toll-like receptor 4 (TLR4), which leads to the production of proinflammatory cytokines and chemokines (*Fassl et al., 2015*). Accordingly, S100A9 knockout (KO) mice exhibit decreased pathogenic outcomes in several mouse models of disease, such as sepsis (*Vogl et al., 2007*), autoimmune disease (*Loser et al., 2010*) or arthritis (*van Lent et al., 2008*). In addition, S100A8 and S100A9 are highly abundant during infectious diseases and exhibit anti-microbial activities. The S100A8/A9 complex plays a crucial role in host defence against bacterial and fungal pathogens by sequestering manganese and zinc ions which compete with high affinity bacterial transporters to import these essential nutrient metals (*Kehl-Fie and Skaar, 2010*; *Kehl-Fie et al., 2011*). In contrast to the proinflammatory role of S100A8/A9, regulatory functions in terms of hyporesponsiveness in phagocytes, resembling a classical endotoxin-induced tolerance, have also been described (*Freise et al., 2019*).

In humans, S100A8/A9 is the most abundant alarmin in many clinically relevant diseases, and is closely associated with disease activity in rheumatoid arthritis (RA), inflammatory bowel disease, sepsis, cardiovascular diseases, multiple sclerosis, acute lung injury (ALI) and psoriasis (*Foell et al., 2004*). Altered S100A8/A9 expression has also been found in different cancer types, including gastric, colorectal, breast, lung, prostate and liver cancer (*Cross et al., 2005*). Despite the high expression in neutrophils and monocytes under inflammatory conditions, and the strong effects of S100A8 and S100A9 on disease activities, transcriptional mechanisms regulating these extreme dynamics of gene expression remain unclear. Identifying the mechanisms regulating *S100a8* and *S100a9* gene expression may open new insights into the pathological processes involving S100A8/A9 during inflammatory conditions.

So far, several potential transcription factors modulating *S100a8* and *S100a9* expression have been described (*Kuruto-Niwa et al., 1998*; *Fujiu et al., 2011*; *Lee et al., 2012*; *Liu et al., 2016*; *Yang et al., 2017*), but their functional relevance remains unresolved. Many of the stated studies used malignant immortalized cell lines or even cell models whose homologous primary cells do not express these genes at all.

To overcome difficulties of artificial expression and malignant cell lines we used ER (estrogen-regulated) Hoxb8 cells, estrogen dependent transiently immortalized myeloid precursor cells that can be differentiated to primary monocytes and granulocytes upon estrogen-withdrawal (*Wang et al., 2006a*), and show the physiologically high dynamics of S100A8 and S100A9 mRNA and protein expression during differentiation. In order to detect genes involved in the regulation of *S100a8* and *S100a9* expression in an unbiased manner, we used a mouse Genome-Scale CRISPR/Cas9 Knockout

(GeCKO) library and screened for monocytes with reduced or absent S100A9 expression. We thereby identified the CCAAT/enhancer-binding protein-family member C/EBPδ as a direct transcriptional regulator of *S100a8* and *S100a9*. Furthermore, we found that the epigenetic factor JMJD3 contributes to *S100a8* and *S100a9* expression in monocytes by erasure of the repressive histone mark H3K27me$_3$ at *S100a8* and *S100a9* promoter regions. Moreover, we confirmed the biomedical relevance of this network a murine model of ALI and in specific monocyte subpopulations in a clinical cohort of patients with cardiovascular disease.

## Results

### Genome-wide CRISPR/Cas9 knockout screen reveals C/EBPδ as a regulatory factor of S100A9 expression

To detect genes involved in the regulation of *S100a8* and *S100a9* expression, we established the mouse GeCKO lentiviral pooled library designed in Cas9-expressing ER-Hoxb8 cells. The used library contained a large mixture of CRISPR sgRNA constructs, where six gRNAs per target gene increase efficiency and enable the analysis of the molecular effects of many thousand genes in one experiment. After infection of Cas9-expressing ER-Hoxb8 precursor cells with CRISPR library lentiviral particles, the cells were differentiated for 3 days in the presence of GM-CSF to induce S100A8 and S100A9 expression. Because we hypothesized that the parallel S100A8 and S100A9 expression is based on a common regulatory mechanism, we assumed that screening of one of the two alarmins was sufficient in the first step. Therefore, cells with no or low S100A9 expression were selected by sorting and considered as hits, whereas the remaining cells functioned as reference cells. To exclude phenotypes that were S100A9$^{low/neg}$ due to general differentiation defects, we pre-gated for CD11b$^+$Ly-6C$^+$ monocytes. DNA of sorted cell pools was purified and analysed by NGS (*Figure 1A*). Intracellular S100A9-FITC staining of precursor and differentiated Cas9 ER-Hoxb8 control monocytes was used as standard for definition of sorting gates. Differentiated Cas9-library ER-Hoxb8 monocytes showed a wider distribution among the gates, indicating the presence of S100A9$^{low/neg}$ expressing cells due to disruptions of regulatory genes caused by CRISPR/Cas9. A small amount of S100A9$^{neg}$ sorted cells were re-analysed by immunoblotting to validate S100A9 deficiency in this cell population (*Figure 1B*). Analysis of CRISPR KO library screen using the Model-based Analysis of Genome-wide CRISPR-Cas9 Knockout (MAGeCK) method (*Li et al., 2014*) resulted in a list of genes for which the respective gRNAs were enriched in the hits sample. The highest number of three gRNAs was found within the top 20 hits targeted *Cebpd*, a gene encoding for a member of the CCAAT/enhancer-binding protein family, C/EBPδ. Other gene hits, namely *Manea*, *Htr1f*, *Ifit3*, *Phf8*, *Hand1*, and *Casp4*, were targeted by two gRNAs (*Figure 1C*). From these, those genes known to be involved in gene regulation, such as the transcription factors *Cebpd* and *Hand1*, as well as the histone demethylase *Phf8*, were selected for validation. Another transcription factor in the top 20 hit list, *Csrp1*, was targeted by only one gRNA but was chosen for validation due to its known functions in gene regulatory mechanisms (*Kamar et al., 2017*). Targeting *Phf8*, *Hand1*, and *Csrp1* only slightly influenced *S100a8* and *S100a9* expression, whereas C/EBPδ deficiency strongly decreased *S100a8* and *S100a9* levels in differentiated monocytes (*Figure 1—figure supplement 1A*). Interestingly, none of the transcription factors previously reported to target *S100a8* and *S100a9* (*Kuruto-Niwa et al., 1998*; *Fujiu et al., 2011*; *Lee et al., 2012*; *Liu et al., 2016*; *Yang et al., 2017*) were found within the hit list of our CRISPR KO library screen (*Figure 1—source data 1*). Nevertheless, to test the published candidate transcription factors ATF3, STAT3, KLF5, IRF7, and C/EBPβ for their effects on *S100a8* and *S100a9* regulation, we created single KO ER-Hoxb8 cell lines of each individual candidate transcription factor. Deficiency of none of the stated candidate transcription factors affects *S100a8* and *S1009* expression during monocyte differentiation, whereas C/EBPδ deficiency had a strong attenuating effect on *S100a8* and *S100a9* expression, as shown on day 2 (*Figure 1—figure supplement 1B*).

### Decreased *S100a8* and *S100a9* expression in C/EBPδ KO monocytes

We confirmed extraordinarily high dynamics of *S100a8* and *S100a9* expression during monocyte/macrophage differentiation. ER-Hoxb8-derived monocytes showed about 590-fold increase in *S100a8* mRNA expression and about 1800-fold increase of *S100a9* mRNA expression on day 2 compared to

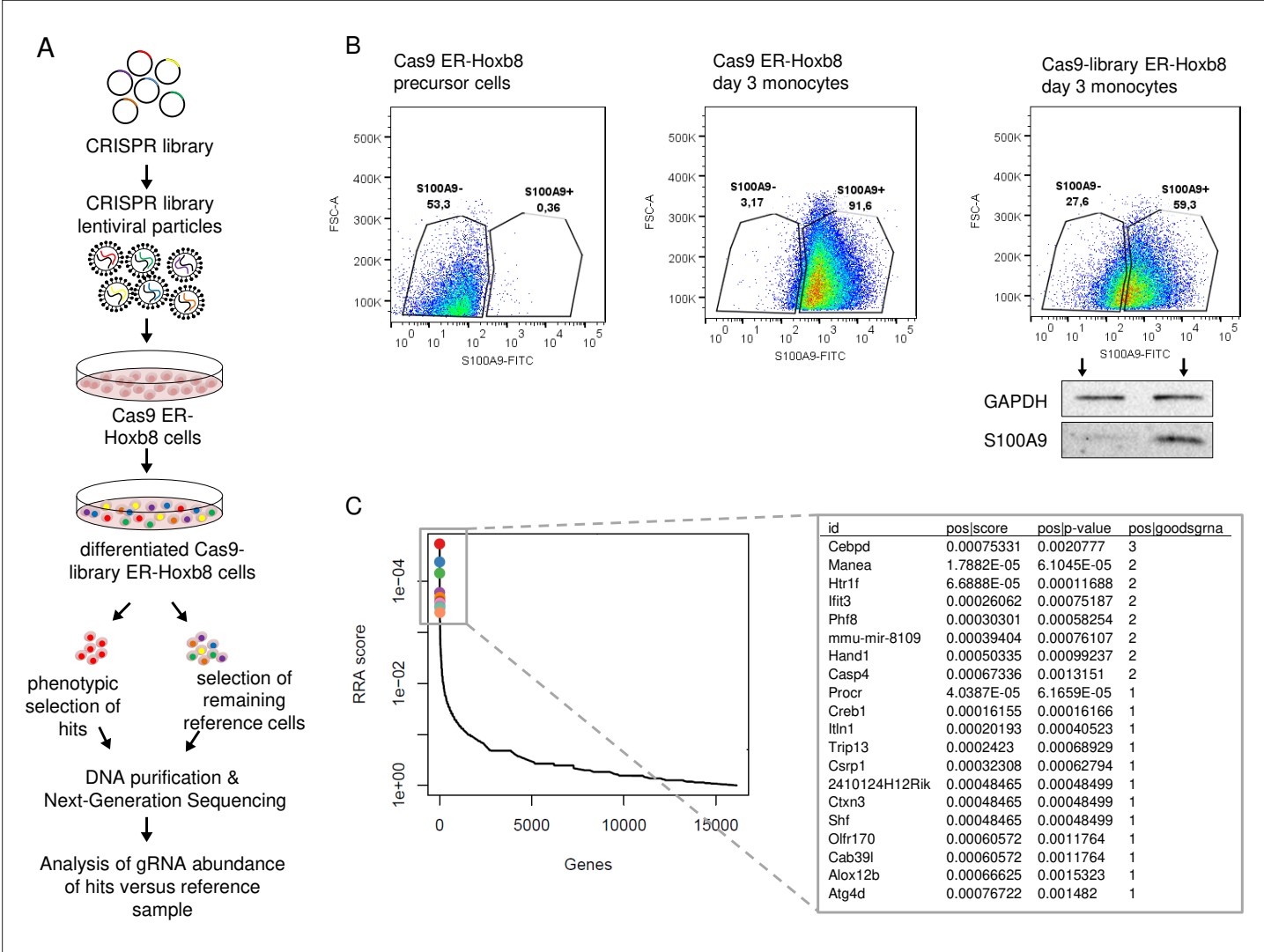

**Figure 1.** Genome-Scale CRISPR Knockout lentiviral pooled library screen to identify S100A9 regulators. (**A**) For genome-wide screen, over 100,000 plasmids, each containing a guide RNA towards different early consecutive exons, were packaged into lentiviral particles. Cas9-expressing ER-Hoxb8 cells were pool-transduced, selected, and differentiated to induce S100A9 expression. Hits and reference cells were collected by sorting according to their phenotypes of interest. DNA of both samples was purified for next-generation sequencing and subsequent analysis. (**B**) Precursor and differentiated Cas9 and Cas9-library ER-Hoxb8 cells were stained intracellularly for S100A9 using a FITC-labelled antibody. Cas9-library ER-Hoxb8 day 3 monocytes with no or lower S100A9 expression were sorted as hits, the remaining cells served as reference cells. (**C**) Data was analysed using the Model-based Analysis of Genome-wide CRISPR-Cas9 Knockout (MAGeCK) software for identification of enriched guide RNAs in the hit sample. Corresponding genes were rank-ordered by robust rank aggregation (RRA) scores. The list states the top 20 genes according to RRA scores, arranged after the number of guides that are enriched in the hit sample. See also *Figure 1—figure supplement 1* and *Figure 1—source data 1*.

The online version of this article includes the following source data and figure supplement(s) for figure 1:

**Source data 1.** Gene summary of Model-based Analysis of Genome-wide CRISPR-Cas9 Knockout (MaGECK) analysis.

**Figure supplement 1.** Relative *S100a8* and *S100a9* expression in differentiated single knockout (KO) ER-Hoxb8 monocytes.

day 0 of differentiation. At day 5 the *S100a8* mRNA expression is already about 70-fold and the *S100a9* mRNA expression about 110-fold downregulated compared to day 2 of differentiation (*Figure 2A, B*).

We confirmed the relevance of C/EBPδ for *S100a8* and *S100a9* expression by creating independent C/EBPδ-deficient ER-Hoxb8 cells from C/EBPδ KO mice. Not only on differentiation day 3, but already at the very beginning of differentiation, when *S100a8* and *S100a9* levels start to rise, C/EBPδ-deficient ER-Hoxb8 monocytes showed significantly reduced levels of both *S100a8* (*Figure 2A*) and *S100a9* (*Figure 2B*) mRNAs compared to wildtype (WT) controls. The same effect was detectable in C/EBPδ-deficient ER-Hoxb8 cells that were differentiated into the neutrophilic lineage (*Figure 2—figure*

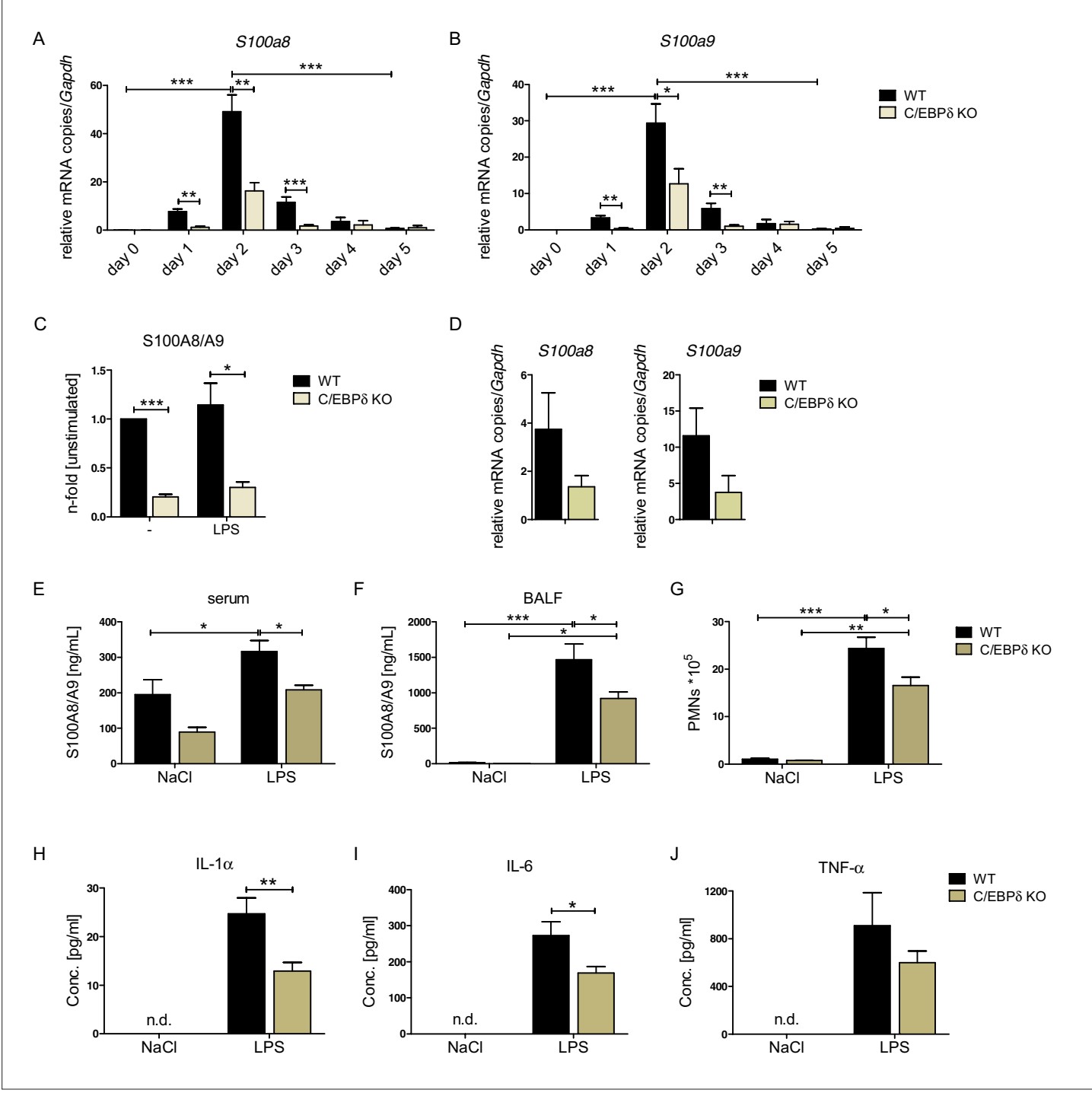

**Figure 2.** S100A8 and S100A9 expression in wildtype (WT) and C/EBPδ knockout (KO) monocytes. (**A**) Relative *S100a8* and (**B**) *S100a9* mRNA level during differentiation of WT and C/EBPδ KO ER-Hoxb8 monocytes were measured using quantitative reverse transcription polymerase chain reaction (qRT-PCR) (n=3–8). (**C**) S100A8/A9 concentrations in supernatant of differentiation day 4 of WT and C/EBPδ KO ER-Hoxb8 monocytes stimulated with 10 ng LPS for 4 hr or left untreated (n=3) were quantified using our in-house mouse S100A8/S100A9 sandwich enzyme-linked immunosorbent assay (ELISA) (n=6). (**D**) Relative *S100a8* and *S100a9* mRNA levels of bone marrow-derived mouse monocytes were measured using qRT-PCR (n=3). (**E**) Serum and (**F**) bronchoalveolar lavage fluid (BALF) of LPS- (WT: n=9, C/EBPδ KO: n=11) or NaCl-only (WT: n=6, C/EBPδ KO: n=3) exposed mice were harvested 4 hr after onset of lung inflammation and were analysed for S100A8/A9 expression, (**G**) for amount of recruited polymorphonuclear leukocytes (PMNs) and for (**H**) IL-1α, (**I**) IL-6, and (**J**) TNF-α production in BALF. Values are the means ± SEM. *p<0.05, **p<0.01, ***p<0.001, by two-tailed Student's t test (A–C, H–J) and by one-way ANOVA with Bonferroni's correction (E–G). See also *Figure 2—figure supplements 1 and 2*.

*Figure 2 continued on next page*

*Figure 2 continued*

The online version of this article includes the following figure supplement(s) for figure 2:

**Figure supplement 1.** *S100a8*, *S100a9*, and *Cebpd* expression kinetics in ER-Hoxb8 cells.

**Figure supplement 2.** Baseline (NaCl) and LPS-induced cytokine production in sera of wildtype (WT) and C/EBPδ knockout (KO) mice.

*supplement 1A*). Accordingly, *Cebpd* and *S100a8* and *S100a9* mRNAs were co-expressed in differentiating WT monocytes and neutrophils, supporting a mechanistic connection (*Figure 2—figure supplement 1B, C*). WT ER-Hoxb8 monocytes secreted significant S100A8/A9 protein amounts, whereas the supernatant of C/EBPδ KO cells contained up to 80% less S100A8/A9 (*Figure 2C*). Accordingly, *S100a8* and *S100a9* expression was decreased upon C/EBPδ deletion in primary bone marrow-derived monocytes (*Figure 2D*). To confirm that C/EBPδ deficiency affects alarmin expression and disease progression in vivo, we used a mouse model for acute lung inflammation. S100A8/A9 levels are locally increased in acute respiratory distress syndrome patients (*Lorenz et al., 2008*) and have been shown to play a role in neutrophil recruitment during ALI in mice (*Chakraborty et al., 2017*). Serum and bronchoalveolar lavage fluid (BALF) from LPS-exposed mice showed highly increased S100A8/A9 levels compared to control mice. In C/EBPδ KO mice, alarmin levels were systemically (serum) and locally (BALF) decreased compared to WT mice at baseline (NaCl) and after LPS exposure (*Figure 2E, F*), confirming our in vitro data. Accordingly, in BALF neutrophil recruitment (*Figure 2G*), as well as in IL-1α and IL-6, cytokine production was significantly decreased in LPS-exposed C/EBPδ KO mice in relation to WT littermates (TNF-α shows a similar trend, *Figure 2H-J*), which highlights the impact of S100A8/A9 on the disease outcome.

## Inflammatory capacities and differentiation kinetics are unaffected in C/EBPδ KO monocytes

Although the proinflammatory molecule S100A8/A9 was strongly reduced in the C/EBPδ KO monocytes, these cells exhibited no general alterations of inflammatory functions, indicating a rather specific effect on *S100a8* and *S100a9* regulation than a general attenuation of inflammatory signaling due to C/EBPδ deficiency. Phagocytosis capacities, examined by using FITC-coupled Latex Beads (*Figure 3A, B*) and by using *Staphylococcus aureus* (*Figure 3C, D*), were even elevated in C/EBPδ KO monocytes. These observations were accompanied by enhanced gene expression of phagocytosis-related PRRs, such as *Ptx3*, *CD209a*, and *Cd14*, in C/EBPδ KO monocytes (*Figure 3—figure supplement 1A, B, D*). ROS production was not influenced by C/EBPδ deficiency (*Figure 3E*). Moreover, analysis of differentiation kinetics revealed no exorbitant differences in the quantities of CD11b$^+$Ly-6C$^+$ between WT and C/EBPδ KO cells, neither during differentiation of bone marrow-derived cells (BMCs) nor of ER-Hoxb8 cells (*Figure 3F*). With regard to monocyte differentiation and polarization capacities, we tested the effect of C/EBPδ deficiency in primary bone marrow-derived mouse monocytes (BMDMs). Whereas expression of M$_1$-monocyte associated markers, such as *Tnfa*, *Il6*, *Inos*, *Cd86*, and *Il1b*, was only slightly decreased in LPS- and IFN-γ-treated C/EBPδ KO monocytes in vitro (*Figure 3—figure supplement 2A*), M$_2$-associated *Il10* expression was significantly decreased upon IL-4-stimulation in C/EBPδ KO cells confirming earlier results (*Liu et al., 2003*). Expression of *Cd163* showed no significant effect in C/EBPδ KO cells (*Figure 3—figure supplement 2B*). Decreased IL-10 expression was also found in sera from LPS-treated and control C/EBPδ KO mice, whereas most other cytokine levels did not differ greatly between WT and C/EBPδ KO mice (*Figure 2—figure supplement 2*).

## Enhanced C/EBPδ expression induces *S100a8* and *S100a9* expression

To test the impact of C/EBPδ induction on *S100* alarmin expression, we infected C/EBPδ-deficient ER-Hoxb8 cells with lentiviral particles carrying a Tet-On system for doxycycline-inducible 3xFlag-C/EBPδ expression (*Figure 4A*). Doxycycline treatment led to expression of the fusion protein 3xFlag-C/EBPδ, as revealed by western blot analysis (*Figure 4B*) and by quantitative reverse transcription polymerase chain reaction (qRT-PCR) in comparison to C/EBPδ-deficient cells (*Figure 4C*). Induction of 3xFlag-C/EBPδ upon doxycycline treatment led to increased *S100a8* and *S100a9* mRNA levels. C*ebpd*, *S100a8*, and *S100a9* mRNA levels in doxycycline-treated TRE_3xFlag-C/EBPδ cells were comparable to WT cells at the same differentiation stage (*Figure 4D*), demonstrating a positive effect of C/EBPδ expression on *S100a8* and *S100a9* regulation. KO of ATF3, a known regulatory attenuator of *Cebpd*

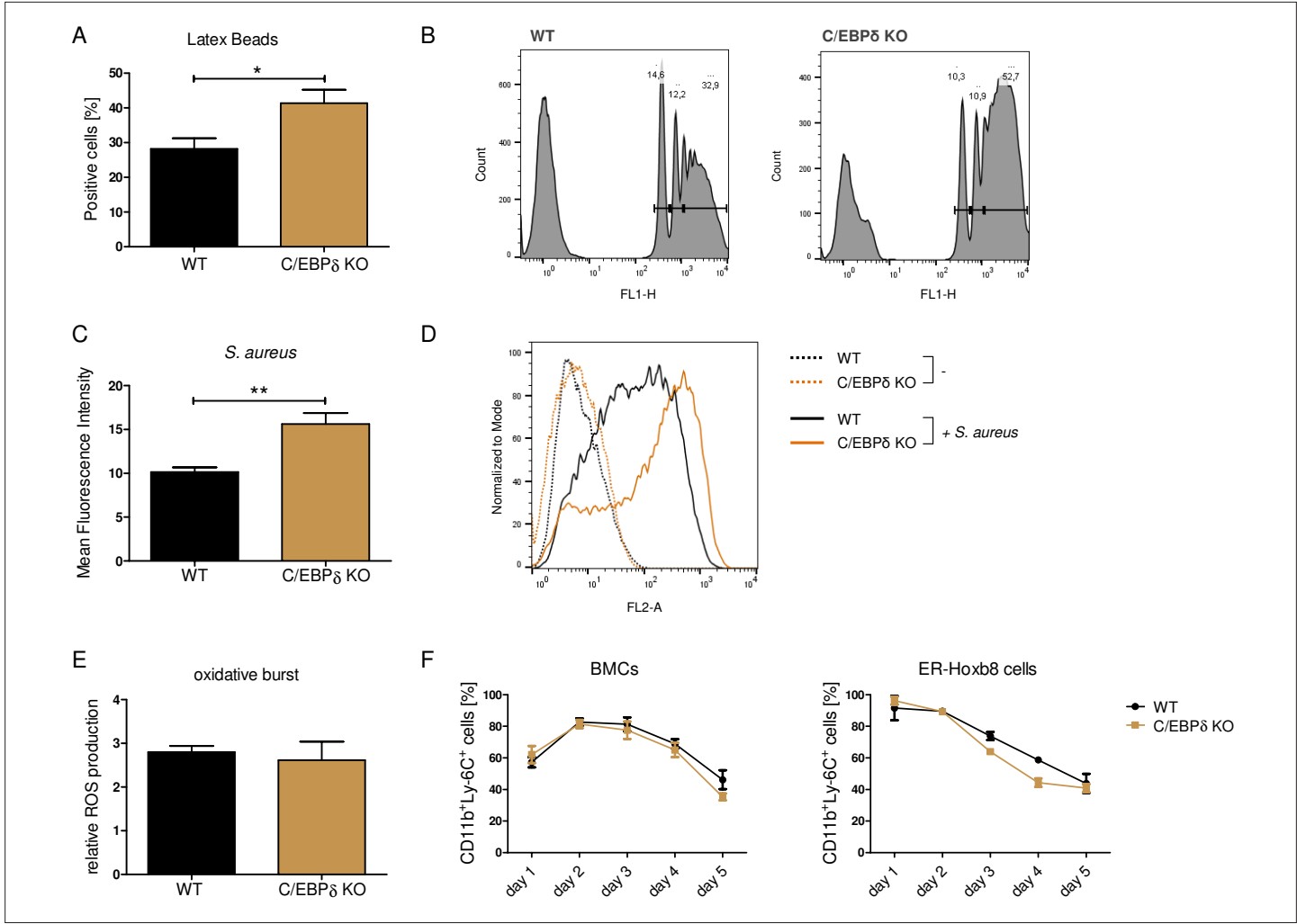

**Figure 3.** Functional properties of wildtype (WT) and C/EBPδ knockout (KO) ER-Hoxb8 monocytes. Differentiated WT and C/EBPδ KO ER-Hoxb8 cells were incubated with green fluorescent latex beads (**A, B**) or with pHrodo Red *Staphylococcus aureus* Bioparticles Conjugate (**C, D**) for 2 hr at a ratio of 10 beads per cells. The phagocytosis was measured using flow cytometry. (**A**) Percentages of phagocytosis of ≥3 beads or (**C**) gMFI shifts of *S. aureus* -treated cells in relation to control cells were used for quantification. Representative histogram plots of Latex Beads- (**B**) and *S. aureus* Bioparticles- (**D**) mediated phagocytosis are presented. (**B**) Histogram plots, showing gates for phagocytosis of one (.), two (...), or ≥3 or more than three (…) beads. (**E**) Differentiated WT and C/EBPδ KO ER-Hoxb8 cells were treated with 10 nM PMA for 15 min and 15 μM DHR123 to measure oxidative burst was measured using DHR123. The fluorescence of the cells was measured using flow cytometry. Relative ROS production as = gMFI shifts of PMA-treated cells in relation to non-treated cells is shown. (**F**) Proportion of CD11b⁺Ly-6C⁺ cells during differentiation of bone marrow cells (BMCs) and ER-Hoxb8 cells was analysed using flow cytometry. Values are the means ± SEM of three to four experiments. *p<0.05, **p<0.01 by two-tailed Student's t test. See also *Figure 3—figure supplements 1 and 2*.

The online version of this article includes the following figure supplement(s) for figure 3:

**Figure supplement 1.** Differential expression of phagocytosis-related genes in wildtype (WT) and C/EBPδ knockout (KO) monocytes.

**Figure supplement 2.** Effect of C/EBPδ deletion on polarization of bone marrow-derived monocytes.

expression (*Litvak et al., 2009*), in ER-Hoxb8 monocytes led to *S100a8* and *S100a9* overexpression (*Figure 4E*, *Figure 1—figure supplement 1B*), especially during early stages of differentiation. ATF3 KO cells showed significantly elevated *cebpd* level, indicating a C/EBPδ-mediated effect on the expression of *S100a8* and *S100a9* (*Figure 4F*). In the next step, we created FBXW7-deficient monocytes. FBXW7 is another well-known attenuator of C/EBPδ expression (*Balamurugan et al., 2013*). Lack of this antagonist resulted in an even higher overexpression of *S100a8* and *S100a9* (*Figure 4G*) with huge increases of *Cebpd* levels (*Figure 4H*).

To confirm the biomedical relevance of the identified molecular network, we analysed the expression of these genes in peripheral blood mononuclear cells (PBMCs) and monocyte subpopulations

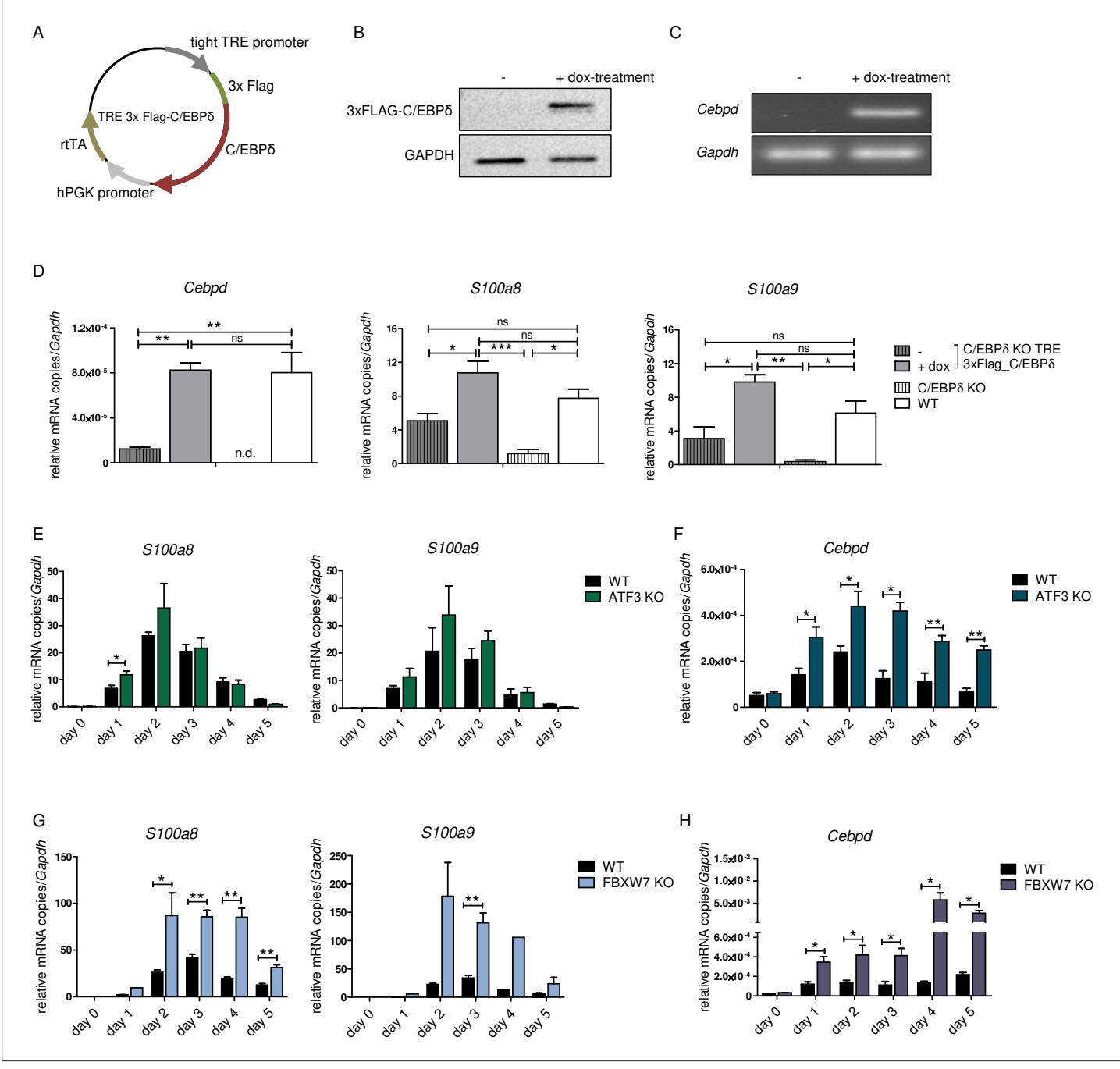

**Figure 4.** *S100a8* and *S100a9* expression in differentiated ER-Hoxb8 cells is dependent on C/EBPδ abundancy. (**A**) Tet-On construct of inducible 3xFlag-C/EBPδ expression due to constitutively expressed rtTA (reverse tetracycline-controlled transactivator) that binds to TRE promoter upon doxycycline treatment was transduced into C/EBPδ knockout (KO) ER-Hoxb8 cells. (**B**) Induction of 3xFlag-C/EBPδ upon doxycycline treatment (2 μg/ml, 24 hr) was analysed by western blot and (**C**) quantitative reverse transcription polymerase chain reaction (qRT-PCR) in comparison to untreated cells. (**D**) Induction of 3xFlag-C/EBPδ was also analysed by qRT-PCR (*Cebpd*), as well as expression of *S100a8* and *S100a9* mRNAs, in untreated and dox-treated C/EBPδ KO TRE_3xFlag-C/EBPδ monocytes and in comparison to wildtype (WT) and C/EBPδ KO monocytes on differentiation day 1 (n=3). (**E, G**) *S100a8* and *S100a9*, (**F, H**) and *Cebpd* mRNA levels were measured using qRT-PCR in precursor and differentiated WT and ATF3 KO (**E, F**) and in WT and FBXW7 KO (**G, H**) ER-Hoxb8 monocytes (n=3–4). Values are the means ± SEM. *p<0.05, **p<0.01, by one-way ANOVA with Bonferroni's correction (**D**) and by two-tailed Student's t test (**E–H**).

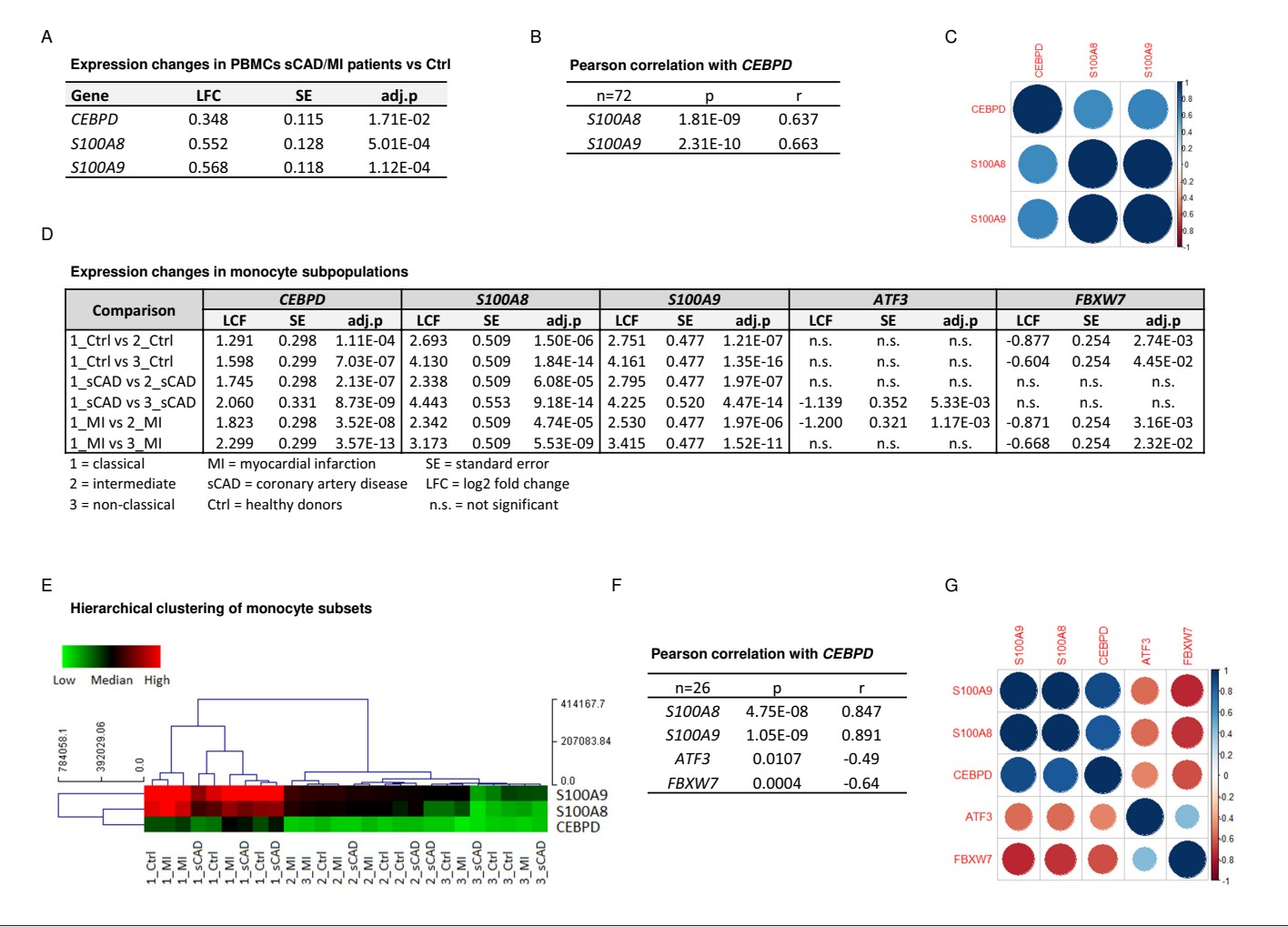

**Figure 5.** *CEBPD* expression positively correlates with *S100A8* and *S100A9* expression in proinflammatory monocytes of myocardial infarction/stable coronary artery disease (MI/sCAD) patients. (**A**) Gene expression changes detected by RNA-sequencing (RNA-seq) in peripheral blood mononuclear cells (PBMCs) of BioNRW participants (n=72, sCAD/MI vs. Ctrl). LFC = log2 fold change, SE = standard error, and adj.p = adjusted p-value. (**B**) Pearson correlation coefficient = r, p-value = p in PBMCs and (**C**) corresponding correlation matrix. (**D**) Gene expression changes of *CEBPD*, *S100A8*, *S100A9*, *ATF3*, and *FBXW7* detected by RNA-seq in monocyte subpopulations of BioNRW participants (n=26, from three individuals in each of the sCAD, MI, and Ctrl diagnostic groups). (**E**) Hierarchical clustering of *S100A8*-, *S100A9*-, and *CEBPD* normalized counts (using Euclidean distance metric with complete linkage). Shown are classical (1), intermediate (2), and non-classical (3) monocytes of healthy donors (Ctrl), MI, and sCAD patients. (**F**) Pearson correlation coefficient = r, p-value = p in monocytes and (**G**) corresponding correlation matrix. See also *Figure 5—source data 1*.

The online version of this article includes the following source data for figure 5:

**Source data 1.** Expression changes in the BioNRW monocytes dataset (RNA-sequencing [RNA-seq], n=26).

of a subset of participants in the BioNRW Study (*Witten et al., 2022*). Here, we found upregulation of *S100A8*, *S100A9*, and *CEBPD* in PBMCs of stable coronary artery disease/myocardial infarction (sCAD/MI) cases, compared against controls (*Figure 5A*), together with a positive correlation of *S100A8* and *S100A9* expression with that of *CEBPD* in these cells (*Figure 5B, C*). Moreover, there was also significant upregulation of these three genes specifically in classical monocytes, compared to intermediate and non-classical monocyte subpopulations (*Figure 5D, E*, for further comparisons, see *Figure 5—source data 1*). A strong positive correlation between the expression of *S100A8* and *S100A9* and *CEBPD* in these monocyte subpopulations was found, suggesting that the expression of these genes is mainly associated with the subset of proinflammatory monocytes. Interestingly, we also found significant albeit milder, negative correlations between the expression of *CEBPD* and its antagonists *FBXW7* and *ATF3* in monocytes (*Figure 5F, G*).

## C/EBPδ-binding sites within *S100a8* and *S100a9* promoter regions

Chromatin immunoprecipitation (ChIP) revealed 3xFlag-C/EBPδ binding on *S100a8* and *S100a9* promoter regions just before or within the predicted enhancers (*Figure 6A*). Co-transfection of HEK293T cells with GFP-expressing *s100* reporter vectors, together with doxycycline-inducible 3xFlag-C/EBPδ vector (TRE_3xFlag-C/EBPδ) or its backbone lacking the 3xFlag-C/EBPδ construct (TRE_ctrl), was performed to further examine specific C/EBPδ binding (*Figure 6B*). Doxycycline treatment resulted in 3xFlag-C/EBPδ protein expression after 24 hr post-transfection in 3xFlag-C/EBPδ vector transfected cells (*Figure 6C*). Transfection of either *S100a8* reporter construct (*Figure 6D*) or *S100a9* reporter construct (*Figure 6F*) together with 3xFlag-C/EBPδ vector led to enhanced GFP expression upon doxycycline treatment in comparison to co-transfection with backbone plasmid (TRE_ctrl). Next, we modified predicted C/EBP-binding sites on *S100* promoters by mutagenesis of the corresponding vectors. Again, co-transfection of mutated *S100* reporter vectors and doxycycline-dependent 3xFlag-C/EBPδ vector was performed to analyse the relevance of specific C/EBPδ-binding sites. Two sites within the *S100a8* promoter region, stated as site 2 and site 3 (*Figure 6E*), and one within the *S100a9* promoter region, stated as site 4 (*Figure 6G*), caused a reduced or absent GFP expression upon co-transfection when deleted. These binding sites, in turn, were located within the *S100a8* and *S100a9* promoter regions where C/EBPδ binding was confirmed by ChIP (*Figure 6A*).

## Epigenetic landscape on *S100* promoter regions reflects *S100a8* and *S100a9* expression in monocytes

Regulation of gene expression relies on variable factors; among these are chromatin structure and epigenetic features. To measure changes in chromatin accessibility in monocytic progenitors and in *S100a8*- and *S100a9*-expressing monocytes, we performed ATAC-seq of precursor and differentiated WT and C/EBPδ KO ER-Hoxb8 cells. This revealed over 1000 regions with differential peaks in all comparisons (*Figure 7A* and *Figure 7—figure supplement 1*). Among the regions with significantly higher ATAC-seq reads in differentiated samples at day 3 of WT cells were the *S100a8* and *S100a9* promoter and enhancer locations. Interestingly, within these regions chromatin accessibility was significantly decreased in C/EBPδ KO in comparison to WT cells at day 3 of differentiation (*Figure 7B, C*). Consistent with the changes in chromatin accessibility at *S100* promoter regions during differentiation, we also found changes in histone marks by ChIP. H3K27 acetylation (H3K27ac), a marker for active transcription, was increased at differentiation day 3 in monocytes over precursor cells at *S100a8* (*Figure 7D*) and *S100a9* loci (*Figure 7E*) in both, WT and C/EBPδ KO cells. In contrast, tri-methylated H3K27 (H3K27me$_3$), associated with gene silencing, was overrepresented in precursor cells over differentiated cells at the same loci in WT cells, whereas H3K27me$_3$ marks did not decrease over the course of differentiation in C/EBPδ KO cells. Accordingly, tri-methylated H3K27 was increased in C/EBPδ KO monocytes, compared to the WT counterparts (*Figure 7F, G*).

## The histone demethylase JMJD3 drives *S100a8* and *S100a9* expression in dependency of C/EBPδ

Decreased *S100a8* and *S100a9* expression in C/EBPδ-deficient day 3 monocytes was mirrored in the epigenetic landscape by only slightly decreased H3K27ac level, but highly increased H3K27me$_3$ level at *S100* promoter regions. Erasure of tri-methylation and di-methylation at H3K27 is known to be catalysed by the histone demethylase JMJD3 (JmjC Domain-Containing Protein 3) (*Xiang et al., 2007*). We found decreased expression of *Jmjd3* in differentiated C/EBPδ KO monocytes, compared to WT cells at the same stage (*Figure 8A*). Further, we used the potent JMJD3 inhibitor GSK-J4 (*Kruidenier et al., 2012*) to block H3K27 demethylation in differentiating ER-Hobx8 cells, and discovered significantly decreased *S100a8* and *S100a9* expression in GSK-J4-treated WT cells (*Figure 8B*). These mRNA quantities were comparable to untreated C/EBPδ-deficient monocytes, whereas the effects on *S100a8* and *S100a9* expression in GSK-J4-treated C/EBPδ-deficient monocytes, compared to the untreated counterparts, were negligible (*Figure 7B*). These effects of GSK-J4 on *S100a8* and *S100a9* expression are in line with increased H3K27me$_3$ marks in GSK-J4-treated WT monocytes, compared to untreated WT cells on both, *S100a8* (*Figure 8C*) and *S100a9* (*Figure 8D*) promoter regions.

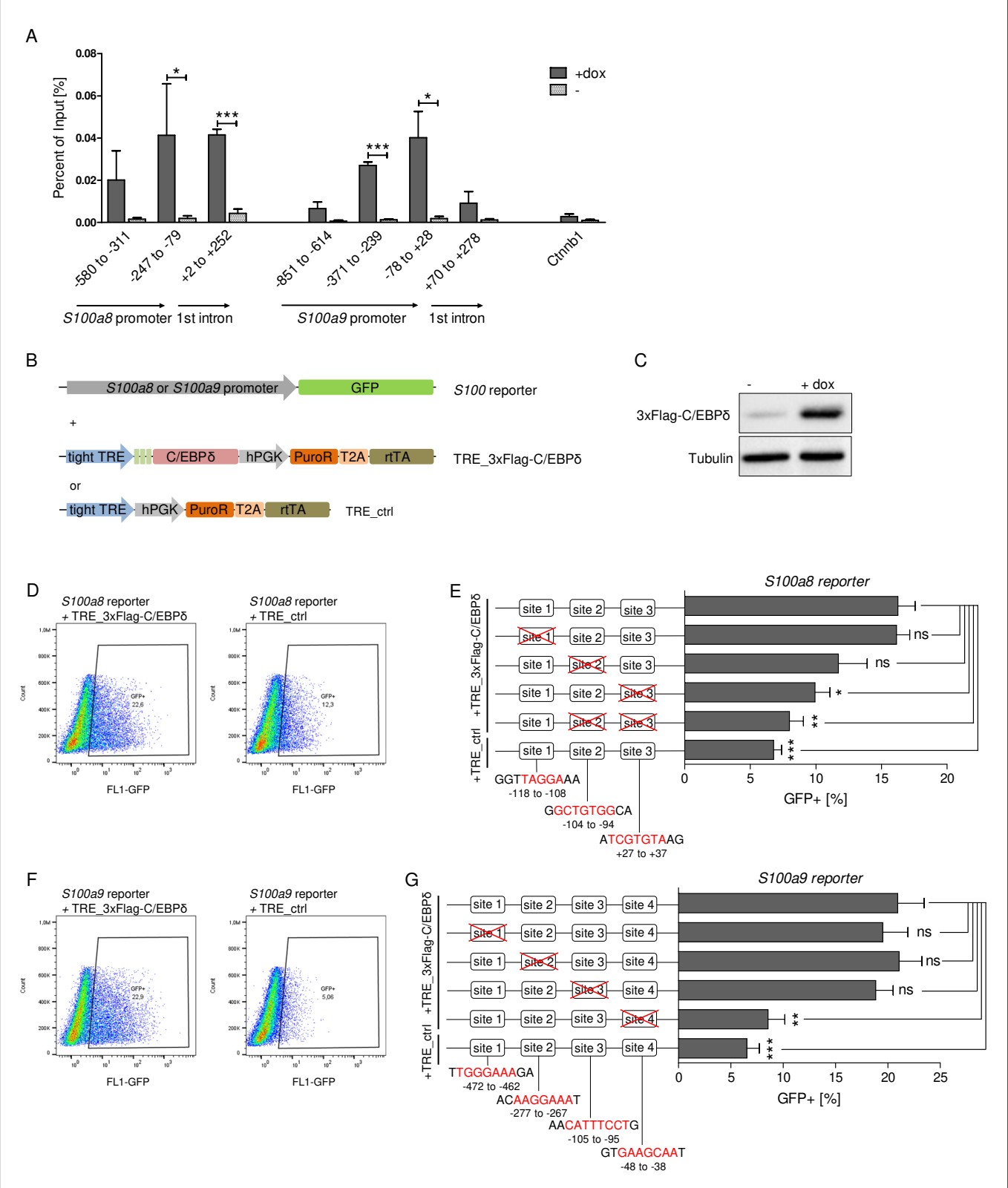

**Figure 6.** C/EBPδ binds to regions within the *S100a8* and *S100a9* promoters. (**A**) Chromatin immunoprecipitation was performed in untreated (-) and dox-treated (+dox) TRE_3xFlag-C/EBPδ monocytes on differentiation day 1 using a Flag-antibody. Purified DNA was analysed using primer pairs flanking different *S100a8* and *S100a9* promoter and intronic regions and a negative control primer pair flanking a random genomic region (n=3). (**B**) Co-transfection of vectors carrying constructs for GFP under the *S100a8* or *S100a9* promoter (reporter), together with the doxycycline-dependent 3xFlag-C/

*Figure 6 continued*

EBPδ expression cassette (TRE_3xFlag-C/EBPδ) or a corresponding control vector lacking the 3xFlag-C/EBPδ expression cassette (TRE_ctrl) in HEK293T cells, was performed. (**C**) Induction of 3xFlag-C/EBPδ upon doxycycline treatment (2 µg/ml, 24 hr) was analysed by western blot. Representative dot plots from flow cytometry analysis show GFP⁺ gates of co-transfected HEK293T cells, either using TRE_3xFlag-C/EBPδ or TRE_ctrl together with *S100a8* reporter (**D**) and with *S100a9* reporter (**F**) upon doxycycline treatment (2 µg/ml, 24 hr). Co-transfection of TRE_3xFlag-C/EBPδ and *S100a8* (**E**) and *S100a9* (**G**) reporter plasmids carrying different mutated possible binding sites was performed, analysed 24 hr post-transfection and compared to co-transfection of TRE_ctrl and *S100* reporter plasmid activities. Suggested C/EBP-binding sites targeted by depletion are indicated by nucleic acids marked in red (n=4–5). Values are the means ± SEM. *p<0.05, **p<0.01, ***p<0.001, ns = not significant, by two-tailed Student's t test.

## Discussion

The ER-Hoxb8 cell system serves as a substitute for murine primary cells of myeloid origin that can be differentiated into phagocytes, such as monocytes and neutrophils. It reflects reliably the differentiation of myeloid progenitor cells to bone marrow-derived monocytes and macrophages. Due to the high expression of S100A8 and S100A9 in the early days of culture around day 3, we describe these stage as monocytes, later days of differentiation with no or low expression of these S100 proteins as macrophages in analogy to the human system. This system allows comparison with in vitro differentiated primary cells (*Wang et al., 2006a*) and, therefore, provides an experimental cell model for analysis of S100A8 and S100A9 expression. Although the alarmins are regarded as key factors in various inflammatory conditions (*Foell and Roth, 2004*), cancer types (*Cross et al., 2005*), and cardiovascular diseases (*Frangogiannis, 2019*), little is known about their transcriptional regulation. The serum concentrations of alarmins correlate with disease severity and activity and, hence, they are reliable biomarkers for monitoring several inflammatory diseases (*Foell et al., 2004*; *Ehrchen et al., 2009*). The expression levels of *S100a8* and *S100a9* differ extremely during myeloid differentiation and the promoters of their genes represent probably one of the most dynamic regulatory elements in the myeloid lineage. Whereas both proteins are completely absent in myeloid precursor cells, they are highly expressed in monocytes and neutrophils, which suggests that highly dynamic regulatory mechanisms drive *S100a8* and *S100a9* expression.

The CRISPR/Cas9-mediated KO screening approach based on a lentiviral pooled library has been used so far to investigate various mechanisms, such as immunity-related pathways and cancer-modulating events (*Kweon and Kim, 2018*). In this study, our unbiased genome-wide screening approach allowed the identification of C/EBPδ as a factor involved in S100A9 regulation during murine monocyte differentiation. We further focused our investigations on *Cebpd* because this gene was in the top list of the robust rank aggregation (RRA) scores and showed the highest numbers of guide RNAs with efficient effects on S100A9 expression in our screening. This redundancy of independent parameters helped to distinguish true positive from false positive hits. Furthermore, a robust phenotype-of-interest, such as a clear S100A9 protein signal at day 3 of monocyte differentiation, allowed reliable negative selection in Cas9-library monocytes. Selection of remaining cells served as a reference control to distinguish true from false positives. The specificity of our selection procedure was confirmed at the protein level by western blot analysis of sorted cell populations. CRISPR/Cas9-based functional genomic screening has been reported to be highly specific, thereby causing fewer cases of false positives in direct comparison with knockdown analysis by RNA interference (*Shalem et al., 2014*). We were now able to identify a novel regulator of S100A8 and S100A9 using this unbiased method. By pre-gating on CD11b⁺Ly-6C⁺ monocytes, we revealed C/EBPδ as a specific and differentiation-independent regulator of S100A8 and S100A9, excluding pathways linked to general functions or development of phagocytes. Re-analysis of S100A9 expression by immunoblotting of sorted cell populations verified the gates set (*Figure 1B*). However, validation of four candidate genes from our hit list demonstrates the limitation of the CRISPR screening system. Not every targeted hit, such as *Phf8*, *Hand1*, and *Csrp1*, showed a prominent effect on *S100a8* and *S100a9* expression. Possibly, the slight differences are rather due to indirect effects on *S100* expression than due to deficiency of actual direct regulators. Another interesting gene within our top 20 hit list is the transcription factor *creb1* (*Figure 1C*) that is known to mediate cytokine signaling at least in human neutrophils and could therefore be tested for *S100a8* and *S100a9* regulation in future experiments (*Mayer et al., 2013*). Known regulators of S100A8 and S100A9 that are also essential for general differentiation, such as PU.1 (*Xu et al., 2021*), were very likely excluded due to our monocyte pre-gating strategy. Our data indicate that this screening technique seems not optimal enough to identify all factors of an

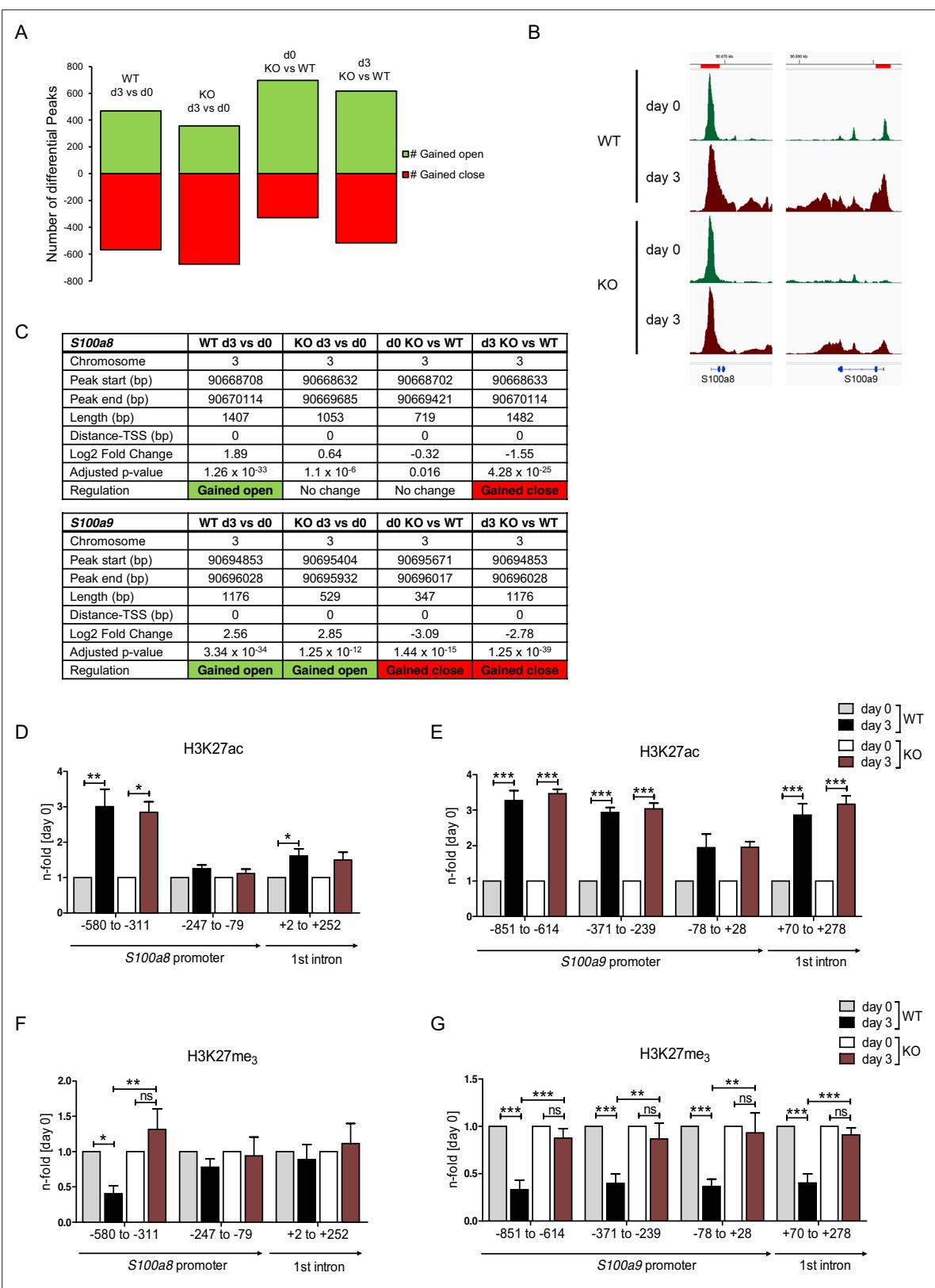

**Figure 7.** Analysis of chromatin accessibility and epigenetic features within *S100a8* and *S100a9* promoter regions. Assay for transposase-accessible chromatin using sequencing (ATAC-seq) was executed in precursor (day 0=d0) and differentiated (day 3=d3) wildtype (WT) and C/EBPδ knockout (KO) (KO) ER-Hoxb8 monocytes. (**A**) Differential peak analysis between all conditions was performed (n=3). (**B**) Combined gene tracks showing ATAC-seq reads of precursor (day 0) and differentiated (day 3) WT and C/EBPδ KO cells at the *S100a8* and *S100a9* gene regions. (**C**) Differential accessibility at

*Figure 7 continued on next page*

*Figure 7 continued*

*S100a8* and *S100a9* loci was analysed for all comparisons. Chromatin immunoprecipitation was performed using anti-H3K27ac (**D, E**), anti-H3K27me3 (**F, G**) in chromatin of precursor (day 0) and differentiated (day 3) WT and C/EBPδ KO (KO) ER-Hoxb8 monocytes. Purified DNA was analysed using primer pairs flanking different *S100a8* (**D, F**) and *S100a9* (**E, G**) promoter regions (n=3–6). N-folds are based on percent of input values of respective day 0 ChIP-PCR samples. Values are the means ± SEM. *p<0.05, **p<0.01, ***p<0.001, ns = not significant, by one-way ANOVA with Bonferroni's correction. See also *Figure 7—figure supplement 1* and *Figure 7—source data 1*.

The online version of this article includes the following source data and figure supplement(s) for figure 7:

**Source data 1.** Differential peak analysis based on assay for transposase-accessible chromatin using sequencing (ATAC-seq) data of wildtype (WT) and C/EBPδ knockout (KO) day 0 and day 3 ER-Hoxb8 cells.

**Figure supplement 1.** Differential peak analysis based on assay for transposase-accessible chromatin using sequencing (ATAC-seq) data of wildtype (WT) and C/EBPδ knockout (KO) day 0 and day 3 ER-Hoxb8 cells.

interrelated regulatory protein complex, but it is sufficient for unbiased detection of single factors for gene regulation, as shown in our study.

We confirmed that the transcription factor C/EBPδ is a direct regulator of S100A8 and S100A9 in murine monocytes using independent approaches. *Cebpd*, and *S100a8* and *S100a9* were co-expressed in differentiating monocytes, and induction of C/EBPδ clearly showed that the expression of *S100a8* and *S100a9* was upregulated by the presence of C/EBPδ. This evidence was further supported by increased *S100a8* and *S100a9* levels caused by deletion of ATF3 and FBXW7, which are natural inhibitors of C/EBPδ. There is a parallel decrease of *Cebpd*, and *S100a8* and *S100a9* expression during later stages of monocyte/macrophage differentiation which may point to a functional relation of C/EBPδ expression and downregulation of *S100* expression as well. However, the late expression kinetics of *S100a8* and *S100a9* in WT and C/EBPδ KO cells clearly point to the presence of additional relevant factors which were not addressed by our screening approach.

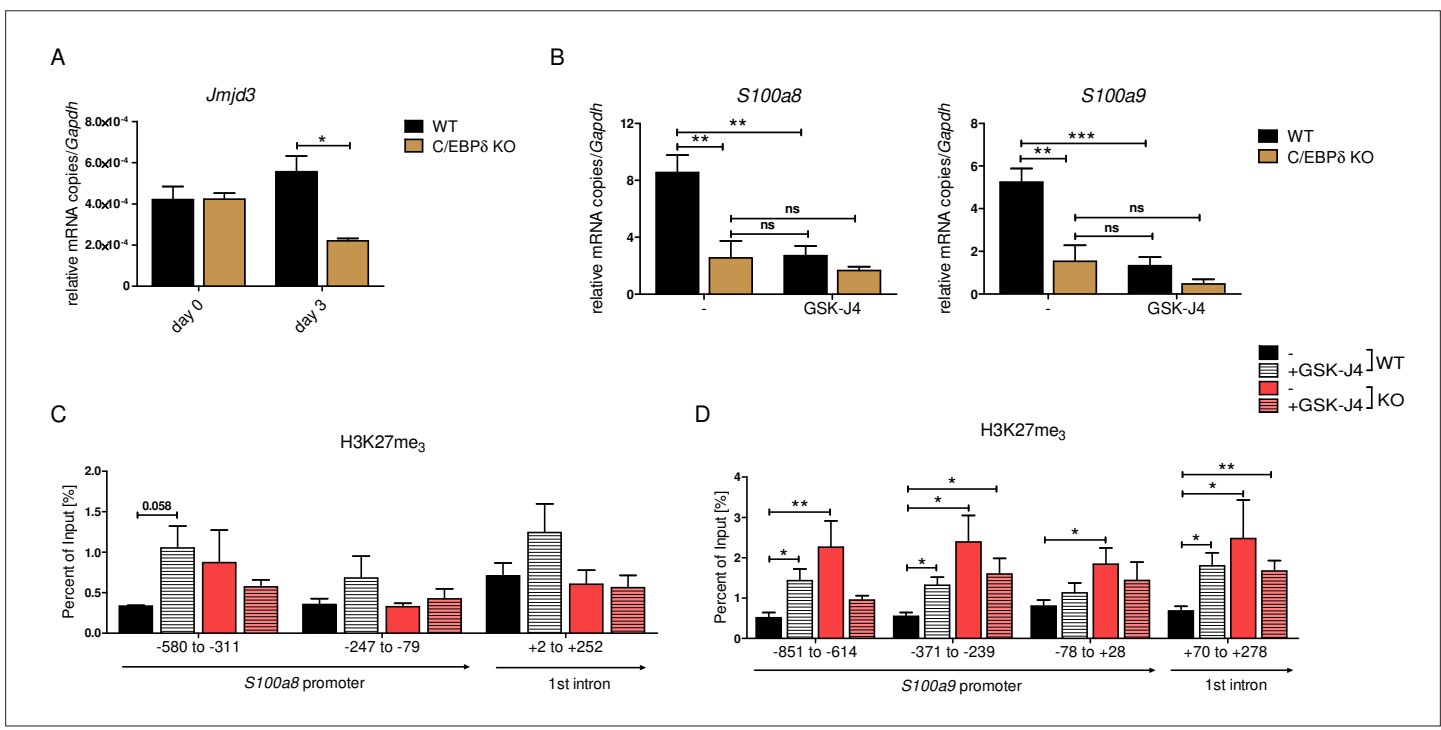

**Figure 8.** JMJD3-mediated demethylation of H3K27me3 is crucial for *S100a8* and *S100a9* expression. (**A**) *Jmjd3* mRNA levels of precursor and differentiated wildtype (WT) and C/EBPδ knockout (KO) ER-Hoxb8 cells were analysed using quantitative reverse transcription polymerase chain reaction (qRT-PCR) (n=3). (**B**) WT and C/EBPδ KO ER-Hoxb8 cells were treated with 5 μM GSK-J4 for 3 days during differentiation and *S100a8* and *S100a9* mRNA levels were analysed using qRT-PCR (n=5). Chromatin immunoprecipitation was performed using anti-H3K27me3 and appropriate IgG control antibodies in chromatin of vehicle controls (-) and treated (+GSK-J4) WT and C/EBPδ KO (KO) ER-Hoxb8 monocytes. Purified DNA was analysed using primer pairs flanking different *S100a8* (**C**) and *S100a9* (**D**) promoter regions (n=3–5). Values are the means ± SEM. *p<0.05, **p<0.01, ***p<0.001, ns = not significant, by one-way ANOVA with Bonferroni's correction (**A, B**) or by two-tailed Student's t test in comparison to WT (-) (**C, D**).

The specificity of our approach was further confirmed by the fact that deficiency of several transcription factors, such as STAT3, KLF5, IRF7, and C/EBPβ, described as S100A8 and S100A9 regulators in previous studies (*Kuruto-Niwa et al., 1998*; *Fujiu et al., 2011*; *Lee et al., 2012*; *Liu et al., 2016*; *Yang et al., 2017*), did neither affect *S100a8* and *S100a9* expression in our ER-Hoxb8 monocytes, nor were these listed as gene hits in our CRISPR/Cas9-mediated KO screening approach. Our ChIP data clearly showed that C/EBPδ specifically binds within *S100a8* and *S100a9* promoter regions. Co-transfection of an inducible C/EBPδ construct and *S100a8* and *S100a9* reporter constructs not only demonstrated *S100* promoter activation due to C/EBPδ expression, but also revealed functional relevance of specific binding sites, via promoter bashing, that are located exactly within the stated promoter regions. The two DNA motifs for specific C/EBPδ responses on the *S100a8* promoter regions did not share the core sequence 5′-C/G GCAAT-3′ that we found within the *S100a9* promoter region in our study. The latter has been described in three other promoters, the human *PPARG2* promoter (*Lai et al., 2008*), the murine and human *CEBPD* promoter itself (*Wang et al., 2021*), and the human *COX2* promoter (*Wang et al., 2006b*). We were able to show that the functionally relevant C/EBPδ-binding sites within the *S100* promoters lie within genome regions which switch from closed to open chromatin states during monocyte differentiation, and concomitant induction of *S100* expression as examined by ATAC-seq.

Our chromatin accessibility data on *S100a8* and *S100a9* promoter regions reflected active *S100a8* and *S100a9* transcription on day 3 cells and accurately mirrored decreased *S100* expression of C/EBPδ KO monocytes in relation to WT counterparts. These observations were again reflected and supported by the characterization of the epigenetic landscape using H3K27ac and H3K27me$_3$ marks. The fact that H3K27me$_3$ marks were strongly decreased in WT monocytes, but not in precursors or C/EBPδ-deficient monocytes, showed the indispensability of H3K27 demethylation for *S100a8* and *S100a9* expression. Moreover, our data demonstrated that the Jumonji C family histone demethylase JMJD3 regulates *S100a8* and *S100a9* expression by erasure of H3K27me$_3$ in dependency of C/EBPδ, which was confirmed by GSK-J4-mediated inhibition of JMJD3 activities. Neither a link of C/EBPδ nor of S100A8/A9 and JMJD3 has been published yet. It has been shown that histone demethylase activities of recombinant JMJD3 on mono-nucleosome substrates is relatively low in contrast to higher activities on bulk histones (*Lan et al., 2007*), suggesting that further factors, such as C/EBPδ, are involved in chromatin binding. Several studies highlight JMJD3 as a regulator of innate immune responses, especially via NF-κB-mediated inflammation in macrophages (*Na et al., 2016*; *Na et al., 2017*; *Davis et al., 2020*). Accordingly, knockdown of JMJD3 affected mainly inflammatory response networks in monocytic THP-1 cells (*Das et al., 2012*) and blocked activation of the NLRP3 inflammasome in bone marrow-derived macrophages (*Huang et al., 2020*). GSK-J4 treatment of mice attenuated disease progression and inflammatory activities in several mouse models for inflammatory diseases, such as arthritis (*Jia et al., 2018*), colitis (*Huang et al., 2020*), and EAE (experimental autoimmune encephalomyelitis) (*Doñas et al., 2016*). Accordingly, GSK-J4 treatment of our ER-Hoxb8 monocytes reduced expression of the proinflammatory alarmins *S100a8* and *S100a9*, which have been shown to drive the inflammatory process of arthritis (*van Lent et al., 2012*). With our study, we have taken a step forward to uncover the role of epigenetic features on *S100a8* and *S100a9* expression and, thereby, on inflammatory conditions in murine monocytes.

We were also able to demonstrate an association of *CEBPD*, and *S100A8* and *S100A9* expression in the context of human cardiovascular disease. The expression of these molecules shows a significant positive correlation not only to each other but also to the manifestation of sCAD and MI in patient-derived PBMCs. Moreover, expression of *CEBPD*, *S100A8* and *S100A9* showed an even stronger association with classical, proinflammatory monocytes (CD14$^{++}$CD16$^{-}$), compared to non-classical (CD14$^{+}$CD16$^{++}$) and intermediate (CD14$^{++}$CD16$^{+}$) monocytes. The endogenous antagonists of C/EBPδ, ATF3 and especially FBXW7, showed a negative correlation of their expression pattern in these monocyte subpopulations. Interestingly, inflammatory monocytes with phagocytic and proteolytic activities have been reported to show an early peak at infarct sites, which are followed by infiltration of non-classical, anti-inflammatory monocytes (*Nahrendorf et al., 2007*; *Dutta and Nahrendorf, 2015*). Genetic deletion of S100A8/A9 was reported to attenuate MI and improve cardiac function in murine models. In contrast, overexpression of *S100a9* in mice increased infarct size and mortality, and treatment with recombinant S100 proteins raised influx of immune cells into the infarct area (*Li et al., 2019*; *Sreejit et al., 2020*). Moreover, serum concentrations of S100A8/A9 are known to be highly

sensitive and prognostic markers for myocardial injury (*Aydin et al., 2019*). Taken together, these data indicate that the C/EBPδ-S100 alarmin axis drives a clinically relevant pathomechanism in cardiovascular disease and probably other inflammation-driven conditions.

There are several published reports suggesting a biomedical relevance of the link between the C/EBPδ and the S100A8/A9 alarmin under other inflammatory conditions as well. For example, C/EBPδ has been shown to play a role in the pathogenesis of psoriasis (*Lan et al., 2020*) and in acute inflammatory signaling by regulating COX-2 (*Wadleigh et al., 2000*), IL-6 (*Litvak et al., 2009*), and TLR4 (*Balamurugan et al., 2013*). Analysis of the genome-wide transcription pattern of monocytes revealed *IL6* as the top gene induced by S100 alarmin stimulation via interaction with TLR4 (*Fassl et al., 2015*), and targeted deletion of S100A9 ameliorated inflammation in a murine psoriasis model (*Zenz et al., 2008*). Additionally, C/EBPδ levels were elevated in mouse models and patients of Alzheimer's disease (AD) (*Li et al., 2004*; *Ko et al., 2012*) and RA (*Nishioka et al., 2000*; *Chang et al., 2012*). In mouse models of AD, downregulation (*Ha et al., 2010*) and deficiency of S100A9 (*Kummer et al., 2012*) had therapeutic effects on disease activity. Also in human studies, S100A9 was found to be associated with AD pathogenesis (*Shepherd et al., 2006*). Beyond that, S100A8 and S100A9 are known key players in the pathogenesis of arthritis in murine models (*van Lent et al., 2012*). Gene expression profiling of blood cells from RA patients receiving anti-TNF-α-based treatment showed that both *CEBPD* and *S100A8* were downregulated by the treatment (*Meugnier et al., 2011*). Uncontrolled activity of S100A8/A9 alarmins drives TNF-induced arthritis in mice (*Vogl et al., 2018*). In the context of human RA, the expression and serum concentrations of S100A8/A9 correlate very well with disease activity and are the first predictive marker for disease relapses in juvenile patients, and of the responses to therapy in juvenile and adult patients (*Moncrieffe et al., 2013*; *Choi et al., 2015*). However, no direct molecular or functional link between S100A8/A9 and C/EBPδ in arthritis has yet been reported. Using an in vivo model for acute lung inflammation, we were now able to show that C/EBPδ-deficient mice express lower S100A8/A9 level than WT littermates in response to LPS exposure, which is accompanied by a milder disease phenotype. In S100A9 KO mice, neutrophil recruitment in the lung was impeded when using the same mouse model (*Chakraborty et al., 2017*), suggesting that S100A8/A9 expression mediates effects of C/EBPδ as a key mediator of LPS-induced lung inflammation (*Yan et al., 2013*). Our data define a new regulatory axis between the transcription factor C/EBPδ and the alarmins S100A8 and S100A9 in myeloid cells which is of relevance in different inflammatory processes and clinical diseases.

## Materials and methods
### Murine model of acute lung inflammation
WT and C/EBPδ KO (kindly provided by Esta Sterneck, National Cancer Institute, Frederick, MD) (*Sterneck et al., 1998*) mice were exposed to LPS from *Salmonella enterica* (0.5 mg/ml, Sigma-Aldrich) in saline or saline (NaCl) only for 45 min inside a cylindrical Pyrex chamber which was connected to a nebulizer. After a 4 hr resting period, mice were anesthetized, blood was collected from heart and BALF was collected for S100A8/A9 measurements by flushing the lungs five times with 0.8 ml NaCl through the trachea. Serum and BALF were saved for later analyses of alarmins and cytokines at –80°C. Migrated polymorphonuclear leukocytes were analysed via Kimura and following Ly-6B.2/Gr-1-antibody (Bio-Rad) staining. Mouse experiments were in accordance with German Animal Welfare Legislation and performed as approved by the North Rhine-Westphalia Office of Nature, Environment and Consumer Protection (LANUV), and the District Government and District Veterinary Office Muenster under the reference number 81-02.04.2019.A445.

### Cell culture
ER-Hoxb8 cells were generated as described earlier (*Wang et al., 2006a*) and grown in RPMI medium (Thermo Fisher Scientific) supplemented with 10% FBS (Biowest), 1% penicillin/streptomycin solution (Sigma-Aldrich), 1% glutamine solution (Thermo Fisher Scientific), 40 ng/ml recombinant mouse GM-CSF (ImmunoTools) an 1 µM β-estradiol (Sigma-Aldrich). For differentiation, precursor cells were washed and incubated in estradiol-free medium containing 40 ng/ml GM-CSF for several days. HEK293T were grown in DMEM (Thermo Fisher Scientific) supplemented with 10% FBS (Biowest) and 1% penicillin/streptomycin solution (Sigma-Aldrich), 1% glutamine solution (Thermo Fisher Scientific),

and 1% sodium pyruvate (Merck). All cell lines were cultured at 37°C, 5% $CO_2$, and routinely screened and found negative for mycoplasma contamination in a PCR-based assay (PromoCell).

## Cell line generation and manipulation

WT, C/EBPδ KO (kindly provided by Esta Sterneck, National Cancer Institute, Frederick, MD) (*Sterneck et al., 1998*) and Cas9-expressing (*Chiou et al., 2015*) ER-Hoxb8 cells originated from corresponding mice. PHF8, CSRP1, HAND1, FBXW7, ATF3, STAT3, KLF5, IRF7, and C/EBPβ KO ER-Hoxb8 cells were generated using CRISPR/Cas9 as described earlier (*Gran et al., 2018*). The oligos for gRNA cloning are listed in *Supplementary file 1a*. For lentiviral production, the lentiGuide-Puro (for GeCKO screen), lentiCRISPRv2-gRNA (for single KO cell lines), or TRE_3xFlag-C/EBPδ was co-transfected into HEK293T cells, together with the packaging plasmids pCMV-VSV-G (AddGene, #8454) and psPAX2 (AddGene, #12260). For transduction of ER-Hoxb8 cells, cells were incubated with lentiviral particles and 8 µg/ml polybrene (Sigma-Aldrich) for 1 hr upon spinfection and selected for several days using puromycin (InvivoGen). For transfection of HEK293T cells, the cells were seeded 1 day prior to transfection. Then, cells were co-transfected with TRE_3xFlag-C/EBPδ and *S100a8* reporter or *S100a9* reporter using the Lipofectamine 3000 Transfection Reagent (Thermo Scientific) according to the manufacturer's manual. For inhibition of JMJD3 activities, cells were treated using 5 µM GSK-J4 HCl (SellekChem) for 3 days. To induce *cebpd* in TRE_3xFlag-C/EBPδ ER-Hoxb8 cells or transfected HEK293T cells, cells were treated using 2 µg/ml doxycycline (Sigma-Aldrich) for 24 hr.

## Isolation of BMDMs

BMDMs were obtained by flushing the femurs from WT and C/EBPδ KO mice. Erythrocytes were depleted by osmotic shock. Cells were washed and separated using a Ficoll gradient (PAN-Biotech). Primary monocytes ($M_0$) were generated by culturing for 3 days in DMEM containing with 10% FBS (Biowest) and 1% penicillin/streptomycin solution (Sigma-Aldrich), 1% glutamine solution (Thermo Fisher Scientific), and 15% of L929 supernatant as a source of macrophage colony-stimulating factor. Stimulation of cells with 50 ng/ml IFN-γ (ImmunoTools) and 10 ng/ml LPS (Sigma-Aldrich) for 24 hr was used for $M_1$-polarization, whereas a 24 hr stimulation with 20 ng/ml IL-4 (Peprotech) led to $M_2$-polarization of BMDMs.

## GeCKO-library screening

Amplification of mouse CRISPR Knockout pooled library (GeCKO v2) in lentiGuide-Puro plasmid, purchased from AddGene (#1000000053) (*Sanjana et al., 2014*), was performed as described (*Joung et al., 2017*). Cas9-expressing ER-Hoxb8 cells, transduced with library lentiviral particles at an MOI of 0.4, were differentiated to day 3. Intracellular S100A9 was stained with an S100A9-FITC coupled antibody using the Foxp3/Transcription Factor Staining Buffer Set (eBioscience). Cells with no/lower S100A9 expression (hits) and cells with normal S100A9 expression (reference) were sorted using an SH800S Cell Sorter (Sony, Minato, Japan) and DNA was purified by phenol-chloroform extraction. Next-generation sequencing was performed as described earlier (*Joung et al., 2017*). Briefly, sgRNA library for next-generation sequencing was prepared via PCR using primers amplifying the target region with Illumina adapter sequences (*Supplementary file 1b*), the purified DNA, and the NEBNext High-Fidelity 2× PCR Master Mix (NEB). PCRs were pooled and purified using the QIAquick PCR Purification Kit (Qiagen). Size and quantity was determined using the Bioanalyzer High Sensitivity DNA Analysis Agilent High Sensitivity DNA Kit (Agilent). Samples were sequenced according to the Illumina user manual with 80 cycles of read 1 (forward) using the NextSeq 500/550 High Output Kit v2.5 (75 Cycles) (Illumina) with the 20% PhiX spike in Illumina PhiX control kit (Illumina).

## Cloning and plasmid production
### TRE_3xFlag-C/EBPδ and TRE_ctrl

The pcDNA 3.1 (-) mouse C/EBPδ expression vector (AddGene, #12559) and annealed oligonucleotides (*Supplementary file 1c*) were digested using *Xba*I and *EcoR*I and then ligated. Using primers carrying restriction enzyme recognition sites (*Supplementary file 1c*), the 3xFlag-C/EBPδ expression cassette was amplified. The resulting amplicon and the pCW57.1 mDux-CA target vector (AddGene, #99284) (*Whiddon et al., 2017*) were digested using *Nhe*I and *Age*I and subsequently ligated. TRE_ctrl was produced by digesting TRE_3xFlag-C/EBPδ using *Nhe*I and *Age*I and by subsequent blunting

of ends by 3' overhang removal and fill-in of 3' recessed (5' overhang) ends using DNA Polymerase I, Large (Klenow) Fragment (NEB) prior to ligation.

### *S100a8* and *S100a9* reporter

To construct *S100* reporter vectors, 1500 bp upstream of *S100a8* and 1800 bp upstream of *S100a9* TSS were amplified from genomic mouse DNA. Using primers carrying restriction enzyme recognition sites (*Supplementary file 1d*), promoter regions were amplified and cloned into pLenti CMV GFP Blast vector (AddGene, #17445) (*Campeau et al., 2009*) using *Xba*I and *Cla*I. Resulting *S100* prom-GFP constructs were cloned into MSCV-PIG-empty vector (AddGene, #105594) (*Xu et al., 2018*) by digestion with *Nsi*I and *Cla*I together with the MSCV backbone to exchange IRES-GFP-cassette with *S100a8/a9*prom-GFP-cassette and subsequent ligation. Proposed C/EBP DNA-binding sites within *S100a8* and *S100a9* promoter regions were identified using the AliBaba2.1 net-based transcription factor-binding site search tool (*Grabe, 2002*), and were mutated by deleting 6–7 base pairs using the QuikChange II XL Site-Directed Mutagenesis Kit (Agilent Technologies). The primers used for mutagenesis are listed in *Supplementary file 1e*. Plasmids were produced in DH5α cells and purified using the PureLinkTM HiPure Plasmid Midiprep Kit (Thermo Scientific).

## qRT-PCR

RNA was isolated using a NucleoSpin Extract II Isolation Kit (Macherey Nagel). The mRNA expression of selected genes was measured by qRT-PCR as described earlier (*Heming et al., 2018*). The primers used are listed in *Supplementary file 1f*. The relative expression level of each target gene was analysed using the $2-\Delta\Delta Cq$ method and was normalized to GAPDH.

## Chromatin immunoprecipitation

For chromatin preparation, progenitor and differentiated ER-Hoxb8 cells were fixed using 1% formaldehyde for 5 min and reaction was stopped by adding 125 mM glycine. Chromatin was extracted as previously described (*Fujita and Fujii, 2013*). Approximately 1–5% of chromatin served as the input sample. DNA from input samples was isolated using phenol-chloroform extraction as described earlier (*Heming et al., 2018*). For immunoprecipitation, 3 µg antibody against Flag (Sigma-Aldrich), H3K27ac (Abcam), H3K27me$_3$ (Cell Signaling Technology), normal Rabbit IgG (Cell Signaling,) or Mouse IgG1, κ Isotype control (BioLegend) was conjugated to 900 µg magnetic Dynabeads-Protein G (Thermo Scientific) at 4°C overnight. Sonicated chromatin was added to AB-conjugated Dynabeads and incubated at 4°C overnight. The Dynabeads were washed as described earlier (*Fujita and Fujii, 2013*). For elution, Dynabeads were incubated twice with elution buffer (0.05 M NaHCO$_3$, 1% SDS) at 65°C for 15 min. DNA from eluates was isolated using phenol-chloroform extraction as with input samples. Values were taken into account only when the amount of DNA pulled down by using the antibody of interest was more than 5-fold increased over DNA pulled down by using IgG antibodies. The primers used for ChIP-PCR are listed in *Supplementary file 1g*.

## ATAC-seq

Precursor and day 3 differentiated WT and C/EBPδ KO ER-Hoxb8 cells were harvested, washed, and cryopreserved in 50% FBS/40% growth media/10% DMSO using a freezing container at –80°C overnight. Cells were shipped to Active Motif to perform ATAC-seq as previously described (*Buenrostro et al., 2013*).

## Measurements of S100A8/A9 protein level

The S100A8/A9 protein concentrations were measured using an in-house S100A8/A9 enzyme-linked immunosorbent assay (ELISA), as previously described (*Vogl et al., 2014*).

## Cytokine measurements by bead-based immunoassay

IL-1α, IFN-γ, GM-CSF, MCP-1, IL-12$_{p70}$, IL-1β, IL-10, IL-6, IL-27, IL-17A, IFN-β, and TNF-α were analysed using the bead-based immunoassay LEGENDplex mouse inflammation panel according to manufacturer's instructions (BioLegend). Analytes were measured by flow cytometry using Navios (Beckmann Coulter).

## Immunoblotting

Cells were lysed in M-PER Mammalian Protein Extraction Reagent (Thermo Scientific) containing a protease inhibitor mixture (Sigma-Aldrich). Protein concentration was determined, and equal amounts (15–30 µg) were run on an SDS-PAGE. After blotting on a nitrocellulose membrane, the membrane was incubated overnight with primary antibodies against: polyclonal rabbit S100A8 and S100A9 antibodies (originated from our own production; *Vogl et al., 2014*), GAPDH (Cell Signaling Technology), α/β-Tubulin (Cell Signaling Technology), and Flag (Sigma-Aldrich). Membranes were incubated with an HRP-linked secondary antibody (Agilent) for 1 hr. Chemiluminescence signal was detected using ChemiDoc XRS+ (Bio-Rad) together with ImageJ (National Institutes of Health) to quantify the signal intensity.

## Phagocytosis

FluoSpheres polystrene microspheres (Thermo Scientific) that were shortly sonicated in a bath sonicator (Latex Beads) or pHrodo Red *S. aureus* Bioparticles Conjugate (*S. aureus* Bioparticles) were added to $5 \times 10^5$ differentiated cells at a ratio 1:10 for 2 hr at 37°C. The rate of phagocytosis was measured by flow cytometry using Navios (Beckmann Coulter).

## Oxidative burst

Cells were stimulated using 10 nM PMA (Abcam) for 15 min or left untreated. After incubation, 15 µM DHR123 (Sigma-Aldrich) were added for another 15 min. The fluorescence signal was analysed using flow cytometry (Navios, Beckmann Coulter).

## FACS analysis

BMCs and ER-Hoxb8 cells were differentiated using 40 ng/ml rmGM-CSF (ImmunoTools), harvested, stained for CD11b, Ly-6C, and appropriate Isotype control antibodies (all BioLegend) and measured using flow cytometry (Navios, Beckmann Coulter) to determine cell population purity and differentiation kinetics.

## RNA-seq

### Study population

For this study, we used bulk mRNA-sequencing (RNA-seq) data of PBMCs and monocytes from two subsets of participants in the German BioNRW Study (*Witten et al., 2022*). BioNRW actively recruits patients undergoing coronary angiography for the diagnosis and percutaneous coronary intervention of coronary artery disease, as well as age and gender matched healthy control individuals without history of cardiovascular disease, all aged 18–70 years of age. Patients receive standard cardiovascular care and medication (ACE-inhibitor, AT1-receptor blocker, β-blocker, diuretics, statin), according to current guidelines. Here, we included a total of 42 patients with sCAD or acute MI, as well as 39 of the corresponding age and sex matched controls.

The BioNRW Study is conducted in accordance with the guidelines of the Declaration of Helsinki. The research protocol, including the case report forms, was approved by the local ethics committee (#245-12). Written informed consent was obtained from all study participants.

### Blood collection and isolation of PBMCs

In case of MI, blood samples were collected during the first 4 days following the event. EDTA blood was drawn from each subject by venipuncture. Sample processing followed within 2 hr. PBMCs were obtained from 40 ml blood by density gradient centrifugation (Ficoll; Biochrom). Lymphocytes were collected and washed twice with PBS. The pellet was re-suspended in freezing medium Cryo-SFM (PromoCell) and cryopreserved.

### Isolation of monocyte subpopulations

After washing, PBMCs were stained with anti-human antibodies specific for CD2 (PE, RPA-2.10, T-cell marker), CD14 (APC, M5E2, monocyte subset differentiation), CD15 (PE, HIM1, granulocyte marker), CD16 (PE-Cy7, 3G8, monocyte subset differentiation), CD19 (PE, HIB19, B-cell marker), CD56 (PE, MY31, NK-cell marker), CD335 (PE, 9E2, NK-cell marker), HLA-DR (FITC, TU36, antigen-presenting

cells) (all from BD Biosciences), as reported by *Cros et al., 2010*. Cells were acquired on a FACS LSR II flow cytometer (BD Biosciences) and analysed using FlowJo software version 10 (Treestar Inc). For sorting of monocyte subsets, PBMCs were stained and sorted on a MoFlo Astrios cell sorter (Beckman Coulter). Cells were sorted in 1 ml of Isol-RNA lysis reagent (5-Prime GmbH) and frozen at –80°C. To avoid gender-specific effects, three representative male samples of each BioNRW diagnostic group (sCAD, MI, and control) were selected to be subjected to cell sorting and subsequent RNA isolation.

## Differential expression analysis in PBMCs and monocytes

For mRNA profiling of PBMCs and monocyte subpopulations using RNA-seq, mRNA was enriched using the NEBNext Poly(A) Magnetic Isolation Module (NEB), followed by cDNA NGS library preparation (NEBNext Ultra RNA Library Prep Kit for Illumina, NEB). The size of the resulting libraries was controlled by the use of a Bioanalyzer High Sensitivity DNA Kit (Agilent Technologies) and quantified using the KAPA Library Quantification Kit for Illumina (Roche). Equimolar, appropriately pooled libraries were sequenced in a single read mode (75 cycles) on a NextSeq500 System (Illumina) using v2 chemistry, yielding in an average QScore distribution of 92%≥Q30 score. They were subsequently demultiplexed and converted to FASTQ files using bcl2fastq v2.20 Conversion software (Illumina). Data was quality controlled using FASTQC software and trimmed for adapter sequences using Trimmomatic (*Bolger et al., 2014*).

## General statistics

The statistical significance of the data was determined using Prism 5.0 software (GraphPad Software, San Diego, CA). Analyses between two groups were performed using an unpaired two-tailed Student's t test. Comparisons among three or more groups were performed by using one-way ANOVA, followed by Bonferroni's multiple means tests for comparing all pairs of columns. Differences were considered statistically significant at a probability (p-value) of <0.05.

## Bioinformatics analysis

### GeCKO-library screening

Analysis of counting the reads for each gRNA and differential analysis was performed using MaGeCK 0.5.9.3, a computational tool to identify important genes from GeCKO-based screens (*Li et al., 2014*). A modified RRA method with a redefined $\rho$ value was used. Former RRA computed a significant p-value for genes in the middle of gRNA ranked list and thereby introducing false positives because the assumption of uniformity is not necessarily satisfied in real applications. Thus, top ranked % gRNAs were selected if their negative binomial p-values were smaller than a threshold, such as 0.05. If j of the n gRNAs targeting a gene were selected, then the modified value is defined as $\rho$ =min (p1, p2,…,pj), where j≤n. This modified RRA method could efficiently remove the effect of insignificant gRNAs in the assessment of gene significance. A permutation test where the gRNAs were randomly assigned to genes was performed to compute a p-value based on the $\rho$ values. By default, 100× ng permutations are performed, where ng is the number of genes. We then compute the FDR from the empirical permutation p-values using the Benjamini-Hochberg procedure.

### ATAC-seq

Sequence analysis was performed by mapping the paired-end 42 bp sequencing reads (PE42) generated by Illumina sequencing (using NextSeq500) to the genome using the BWA algorithm with default settings ('bwa mem'). Only reads that passed Illumina's purity filter, aligned with no more than two mismatches, and mapped uniquely to the genome were used in the subsequent analysis. In addition, duplicate reads ('PCR duplicates') were removed. BAM files provided by Active Motif were used to perform peak calling with MACS2, using paired-end mode with a bandwidth of 200 and q-value cutoff of 0.01. BedGraph files were converted to BigWig format for visualization with the Integrative Genomics Viewer (IGV) (*Thorvaldsdóttir et al., 2013*) using bedGraphToBigWig. Differential accessibility analyses for all possible comparisons between WT and/or C/EBPδ KO in day 0 and day 3 groups were carried out with GUAVA v1 (*Divate and Cheung, 2018*), which implements DESeq2 (*Love et al., 2014*). Differential peaks were defined as those merged intervals within a window of 5000 bp upstream and 3000 bp downstream of the transcription start site (TSS) showing an adjusted p-value (padj) <0.05 and log2 fold change ≥±1.5.

## RNA-seq

The resulting reads were mapped to the human reference genome builds hg19 (monocytes) or hg38 (PBMCs) using Tophat2 (*Kim et al., 2013*) or HISAT2 v2.1.0 (*Kim et al., 2019*), counted by using the R package GenomicAlignments (*Lawrence et al., 2013*) or HTSeq v0.11.2 (*Anders et al., 2015*), and followed by differential expression analysis using DEseq2 (*Love et al., 2014*). The PBMCs dataset used for analysis consisted of 72 individuals, from which 36 were sCAD/MI cases and 36 were controls (21 females and 15 males in each group, mean age: 50.8±12.3 years), while the monocytes dataset contained read counts of classical, intermediate, and non-classical monocyte subpopulations from 9 male individuals (3 MI, 3 sCAD, and 3 controls). One sCAD non-classical monocyte sample had to be excluded from analysis due to low mapping rate; therefore, the monocytes dataset used for analysis contained 26 samples. Genes were considered differentially expressed at adjusted $p<0.05$ (Benjamini-Hochberg method). R was used to perform Pearson correlation tests and generate plots for the genes of interest from the normalized count data.

## Conclusion

We found that the transcription factor C/EBPδ drives expression of the abundant alarmins S100A8 and S100A9, and demonstrated that C/EBPδ binding to specific sites on *S100a8* and *S100a9* promoter regions also induced changes in chromatin accessibility via JMJD3-mediated demethylation of H3K27me$_3$ marks, which includes a so far unknown link. Due to the high relevance of S100A8/A9 alarmin expression in many inflammatory diseases, our findings may point to novel molecular targets for innovative anti-inflammatory therapeutic approaches.

## Acknowledgements

The authors thank Ursula Nordhues, Eva Nattkemper, Heike Hater, Heike Berheide, Sina Mersmann, Elvira Barg, and Marianne Jansen-Rust for their excellent technical support, and Esta Sterneck (Center for Cancer Research, National Cancer Institute, Frederick, MD) for providing the C/EBPδ KO mice. This work was supported by grants from the Interdisciplinary Center of Clinical Research at the University of Münster (Ro2/023/19, Vo2/011/19), the German Research Foundation CRC 1009 B8, B9, and Z2, CRU 342 P3 and P5 and RO 1190/14–1 (to J Roth and T Vogl) and by the EU EFRE Bio NRW programme (005-1007-0006) to MS. The funders had no role in the study design, data collection and analysis, decision to publish, or preparation of the manuscript.

## Additional information

### Funding

| Funder | Grant reference number | Author |
|---|---|---|
| Deutsche Forschungsgemeinschaft | CRC 1009 B9 | Johannes Roth |
| Deutsche Forschungsgemeinschaft | CRC 1009 Z2 | Johannes Roth |
| Deutsche Forschungsgemeinschaft | CRC 1009 B8 | Thomas Vogl |
| Deutsche Forschungsgemeinschaft | CRU 342 P3 | Johannes Roth |
| Deutsche Forschungsgemeinschaft | RO 1190/14-1 | Johannes Roth |
| Deutsche Forschungsgemeinschaft | CRU 342 P5 | Thomas Vogl |
| Interdisciplinary Center of Clinical Research at the University of Münster | Ro2/023/19 | Johannes Roth |

| Funder | Grant reference number | Author |
|---|---|---|
| Interdisciplinary Center of Clinical Research at the University of Münster | Vo2/011/19 | Thomas Vogl |
| EU EFRE Bio NRW programme | 005-1007-0006 | Monika Stoll |

The funders had no role in study design, data collection and interpretation, or the decision to submit the work for publication.

## Author contributions
Saskia-Larissa Jauch-Speer, Conceptualization, Formal analysis, Investigation, Writing - original draft, Writing - review and editing; Marisol Herrera-Rivero, Data curation, Software, Writing - review and editing; Nadine Ludwig, Investigation, Methodology; Bruna Caroline Véras De Carvalho, Jonas Wolf, Investigation; Leonie Martens, Data curation, Software; Achmet Imam Chasan, Methodology; Anika Witten, Supervision; Birgit Markus, Bernhard Schieffer, Resources; Thomas Vogl, Funding acquisition; Jan Rossaint, Conceptualization, Resources; Monika Stoll, Funding acquisition, Project administration, Resources, Supervision; Johannes Roth, Conceptualization, Funding acquisition, Project administration, Supervision, Writing - review and editing; Olesja Fehler, Conceptualization, Investigation, Project administration, Supervision, Writing - review and editing

## Author ORCIDs
Saskia-Larissa Jauch-Speer http://orcid.org/0000-0002-6739-3051
Marisol Herrera-Rivero http://orcid.org/0000-0001-7064-9487
Achmet Imam Chasan http://orcid.org/0000-0001-5137-6890
Johannes Roth http://orcid.org/0000-0001-7035-8348
Olesja Fehler http://orcid.org/0000-0001-6386-7080

## Ethics
Human subjects: The BioNRW Study is conducted in accordance with the guidelines of the Declaration of Helsinki. The research protocol, including the case report forms, was approved by the local ethics committee (#245-12). Written informed consent was obtained from all study participants.

Mouse experiments were in accordance with German Animal Welfare Legislation and performed as approved by the North Rhine-Westphalia Office of Nature, Environment and Consumer Protection (LANUV) and the District Government and District Veterinary Office Muenster under the reference number 81-02.04.2019.A445.

## Decision letter and Author response
Decision letter https://doi.org/10.7554/eLife.75594.sa1
Author response https://doi.org/10.7554/eLife.75594.sa2

# Additional files

## Supplementary files
• Supplementary file 1. List of oligonucleotides used for cloning, (qRT)PCR and mutagenesis. (a) List of guides (stated in 5'–3' orientation) for cloning into lentiCRISPR v2, related to Materials and methods. (b) List of primer (stated in 5'–3' orientation) for amplifying GeCKO library and NGS, related to Materials and methods. (c) List of oligonucleotides (stated in 5'–3' orientation, fw: forward, rv: reverse) for cloning steps to construct TRE_3xFlag-C/EBPδ vector, related to Materials and methods. (d) List of oligonucleotides (stated in 5'–3' orientation, fw: forward, rv: reverse) for cloning steps to construct *S100a8* and *S100a9* reporter construct, related to Materials and methods. (e) List of oligonucleotides (stated in 5'–3' orientation) for mutagenesis to disrupt specific sites within *S100a8* and *S100a9* reporter vectors, related to Materials and methods. (f) List of quantitative reverse transcription polymerase chain reaction (qRT-PCR) primer (in 5'–3' orientation) used for qRT-PCR, related to Materials and methods. (g) List of chromatin immunoprecipitation (ChIP)-PCR primer (in 5'–3' orientation) for *S100a8* and *S100a9* genomic locations, related to Materials and methods.

• Transparent reporting form

## Data availability

GeCKO-screen data are deposited in ID PRJNA754262 at the NCBI's Database, RNA-seq data are deposited in ID 706411 at the NCBI SRA repository and ATAC-seq data are deposited in ID GSE200730 at the NCBI's GEO data repository.

The following datasets were generated:

| Author(s) | Year | Dataset title | Dataset URL | Database and Identifier |
|---|---|---|---|---|
| Jauch-Speer SL, Herrera-Rivero M, Ludwig N, Véras De Carvalho BC, Martens L, Wolf J, Imam Chasan A, Witten A, Markus B, Schieffer B, Vogl T, Roissant J, Stoll M, Roth J, Fehler O | 2021 | GeCKO screen | https://www.ncbi.nlm.nih.gov/bioproject/PRJNA754262 | NCBI BioProject, PRJNA754262 |
| Witten A, Martens L, Schäfer A-C, Troidl C, Pankuweit S, Vlacil A-K, Oberoi R, Schieffer B, Grote K, Stoll M, Markus B | 2021 | monocyte subpopulation profiling | https://www.ncbi.nlm.nih.gov/bioproject/PRJNA706411 | NCBI BioProject, PRJNA706411 |
| Jauch-Speer SL, Herrera-Rivero M, Ludwig N, Véras De Carvalho BC, Martens L, Wolf J, Imam Chasan A, Witten A, Markus B, Schieffer B, Vogl T, Rossaint J, Stoll M, Roth J, Fehler O | 2022 | ATAC-seq in precursor and differentiated ER-Hoxb8 cells | https://www.ncbi.nlm.nih.gov/geo/query/acc.cgi?acc=GSE200730 | NCBI Gene Expression Omnibus, GSE200730 |

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

# Appendix 1

## Appendix 1—key resources table

| Reagent type (species) or resource | Designation | Source or reference | Identifiers | Additional information |
|---|---|---|---|---|
| gene (*Mus musculus*) | *S100a8* | NCBI | Gene ID:20201 | |
| gene (*Mus musculus*) | *S100a9* | NCBI | Gene ID:20202 | |
| Strain, strain background (*Escherichia coli*) | Subcloning Efficiency DH5α Competent Cells | Thermo Fisher Scientific | Cat# 18265017 | |
| Strain, strain background (*Staphylococcus aureus*) | pHrodo Red *S. aureus* Bioparticles Conjugate | Thermo Fisher Scientific | Cat# A10010 | $5 \times 10^6$ per $0.5 \times 10^6$ cells |
| Genetic reagent (*Mus musculus*) | Cebpd$^{tm1Pfj}$/Cebpd$^{tm1Pfj}$ | PMID:9724803 | RRID: MGI:3606309 | Esta Sterneck (CCR, NIH, MD) |
| Genetic reagent (*Mus musculus*) | B6J.129(Cg)-Igs2$^{tm1.1(CAG-cas9*)Mmw}$/J | Jackson Laboratory | Stock# 028239 RRID:IMSR_JAX:028239 | PMID:26178787 |
| Cell line (*Mus musculus*) | WT ER-Hoxb8 cells | This paper | N/A | AG Roth (Institute of Immunology, WWU Münster) |
| Cell line (*Mus musculus*) | Cas9-expressing ER-Hoxb8 cells | This paper | N/A | AG Roth (Institute of Immunology, WWU Münster) |
| Cell line (*Mus musculus*) | C/EBPδ KO ER-Hoxb8 cells | This paper | N/A | AG Roth (Institute of Immunology, WWU Münster) |
| Cell line (*Homo sapiens*) | HEK293T | ATCC | RRID:CVCL_0063 | |
| Transfected construct (*synthetic*) | Cas9+PHF8 gRNA in WT ER-Hoxb8 cells | This paper | N/A | AG Roth (Institute of Immunology, WWU Münster) |
| Transfected construct (*synthetic*) | Cas9+CSRP1 gRNA in WT ER-Hoxb8 cells | This paper | N/A | AG Roth (Institute of Immunology, WWU Münster) |
| Transfected construct (*synthetic*) | Cas9+HAND1 gRNA in WT ER-Hoxb8 cells | This paper | N/A | AG Roth (Institute of Immunology, WWU Münster) |
| Transfected construct (*synthetic*) | Cas9+FBXW7 gRNA in WT ER-Hoxb8 cells | This paper | N/A | AG Roth (Institute of Immunology, WWU Münster) |
| Transfected construct (*synthetic*) | Cas9+ATF3 gRNA in WT ER-Hoxb8 cells | This paper | N/A | AG Roth (Institute of Immunology, WWU Münster) |
| Transfected construct (*synthetic*) | Cas9+STAT3 gRNA in WT ER-Hoxb8 cells | This paper | N/A | AG Roth (Institute of Immunology, WWU Münster) |
| Transfected construct (*synthetic*) | Cas9+KLF5 gRNA in WT ER-Hoxb8 cells | This paper | N/A | AG Roth (Institute of Immunology, WWU Münster) |
| Transfected construct (*synthetic*) | Cas9+IRF7 gRNA in WT ER-Hoxb8 cells | This paper | N/A | AG Roth (Institute of Immunology, WWU Münster) |
| Transfected construct (*synthetic*) | Cas9+C/EBPβ gRNA in WT ER-Hoxb8 cells | This paper | N/A | AG Roth (Institute of Immunology, WWU Münster) |
| Transfected construct (*synthetic*) | MSCV *S100a8* prom_GFP in HEK293T | This paper | N/A | AG Roth (Institute of Immunology, WWU Münster) |
| Transfected construct (*synthetic*) | MSCV *S100a8* s1 prom_GFP in HEK293T | This paper | N/A | AG Roth (Institute of Immunology, WWU Münster) |
| Transfected construct (*synthetic*) | MSCV *S100a8* s2 prom_GFP in HEK293T | This paper | N/A | AG Roth (Institute of Immunology, WWU Münster) |
| Transfected construct (*synthetic*) | MSCV *S100a8* s3 prom_GFP in HEK293T | This paper | N/A | AG Roth (Institute of Immunology, WWU Münster) |
| Transfected construct (*synthetic*) | MSCV *S100a8* s2+3 prom_GFP in HEK293T | This paper | N/A | AG Roth (Institute of Immunology, WWU Münster) |
| Transfected construct (*synthetic*) | MSCV *S100a9* prom_GFP in HEK293T | This paper | N/A | AG Roth (Institute of Immunology, WWU Münster) |
| Transfected construct (*synthetic*) | MSCV *S100a9* s1 prom_GFP in HEK293T | This paper | N/A | AG Roth (Institute of Immunology, WWU Münster) |
| Transfected construct (*synthetic*) | MSCV *S100a9* s2 prom_GFP in HEK293T | This paper | N/A | AG Roth (Institute of Immunology, WWU Münster) |
| Transfected construct (*synthetic*) | MSCV *S100a9* s3 prom_GFP in HEK293T | This paper | N/A | AG Roth (Institute of Immunology, WWU Münster) |

*Appendix 1 Continued on next page*

*Appendix 1 Continued*

| Reagent type (species) or resource | Designation | Source or reference | Identifiers | Additional information |
|---|---|---|---|---|
| Transfected construct (*synthetic*) | MSCV *S100a9* s4 prom_GFP in HEK293T | This paper | N/A | AG Roth (Institute of Immunology, WWU Münster) |
| Transfected construct (*synthetic*) | TRE_3xFlag-C/EBPδ in HEK293T | This paper | N/A | AG Roth (Institute of Immunology, WWU Münster) |
| Transfected construct (*synthetic*) | TRE_ctrl in HEK293T | This paper | N/A | AG Roth (Institute of Immunology, WWU Münster) |
| Biological sample (*Homo sapiens*) | Patient-derived PBMCs | PMID:35379829 | N/A | BioNRW Study |
| Antibody | Anti-mouse S100A8 (rabbit polyclonal) | PMID:25098555 | N/A | ELISA (1:460) WB (1:500) |
| Antibody | Anti-mouse S100A9 (rabbit polyclonal) | PMID:25098555 | N/A | ELISA (1:2300) WB (1:1000) |
| Antibody | Anti-mouse S100A9-FITC (rabbit polyclonal) | This paper | N/A | FACS (5 μg/ml) |
| Antibody | Anti-mouse Ly-6B.2 Alloantigen (7/4) (rat monoclonal) | BioRad | Cat# MCA771F, RRID: AB_322951 | Flow cytometry (1:200) |
| Antibody | Anti-mouse Gr-1 (RB6-8C5) from hybridoma cells (recombinant monoclonal) | This paper | N/A | Flow cytometry (1:200) AG Zarbock (Department of Anesthesiology, UKM) |
| Antibody | Anti-mouse CD11b-Pb (M1/70) (rat monoclonal) | BioLegend | Cat# 101224, RRID: AB_755986 | FACS (1:100) Flow cytometry (1:200) |
| Antibody | Anti-mouse Ly-6C-PE (HK1.4) (rat monoclonal) | BioLegend | Cat# 128008, RRID: AB_1186132 | FACS (1:100) Flow cytometry (1:200) |
| Antibody | Anti-IgG2b, κ Isotype control-Pb(RTK4530) (rat monoclonal) | BioLegend | Cat# 400627, RRID: AB_493561 | Flow cytometry (1:200) |
| Antibody | Anti-IgG2c, κ Isotype control-PE (RTK4174) (rat monoclonal) | BioLegend | Cat# 400707, RRID: AB_326573 | Flow cytometry (1:200) |
| Antibody | Anti-GAPDH (14 C10) (rabbit monoclonal) | Cell Signaling Technology | Cat# 2118, RRID: AB_561053 | WB (1:1000) |
| Antibody | Anti-alpha/beta-Tubulin (rabbit polyclonal) | Cell Signaling Technology | Cat# 2148, RRID: AB_2288042 | WB (1:1000) |
| Antibody | Anti-H3K27me$_3$ (C36B11) (rabbit monoclonal) | Cell Signaling Technology | Cat# 9733, RRID: AB_2616029 | ChIP(3 μg) |
| Antibody | Normal IgG control (rabbit polyclonal) | Cell Signaling Technology | Cat# 2729, RRID: AB_1031062 | ChIP(3 μg) |
| Antibody | Anti-H3K27ac (rabbit polyclonal) | Abcam | Cat# ab4729, RRID: AB_2118291 | ChIP(3 μg) |
| Antibody | Anti-FLAG M2 (mouse monoclonal) | Sigma-Aldrich | Cat# F1804, RRID: AB_262044 | ChIP(3 μg) WB (1:1500) |
| Antibody | IgG1 Isotype control (MOPC-21) (mouse monoclonal) | BD Biosciences | Cat# 554121, RRID: AB_395252 | ChIP(3 μg) |
| Antibody | Anti-Rabbit Immunoglobulins/HRP (goat polyclonal) | Agilent | Cat# P0448, RRID: AB_2617138 | WB (1:4000) |
| Antibody | Anti-Mouse Immunoglobulins/HRP (rabbit polyclonal) | Agilent | Cat# P0260, RRID: AB_2636929 | WB (1:2000) |
| Antibody | Anti-CD2-PE (RPA-2.10) (mouse monoclonal) | BD Biosciences | Cat# 555327, RRID: AB_395734 | FACS (1:20) |
| Antibody | Anti-CD14-APC (M5E2) (mouse monoclonal) | BD Biosciences | Cat# 555399, RRID: AB_398596 | FACS (1:20) |
| Antibody | Anti-CD15-PE (HI98) (mouse monoclonal) | BD Biosciences | Cat# 555402, RRID: AB_395802 | FACS (1:5) |
| Antibody | Anti-CD19-PE (HIB19) (mouse monoclonal) | BD Biosciences | Cat# 555413, RRID: AB_395813 | FACS (1:20) |
| Antibody | Anti-CD16-PE-Cy7 (3 G8) (mouse monoclonal) | BD Biosciences | Cat# 557744, RRID: AB_396850 | FACS (1:20) |

*Appendix 1 Continued on next page*

*Appendix 1 Continued*

| Reagent type (species) or resource | Designation | Source or reference | Identifiers | Additional information |
| --- | --- | --- | --- | --- |
| Antibody | Anti-CD56-PE (MY31) (mouse monoclonal) | BD Biosciences | Cat# 556647, RRID: AB_396511 | FACS (1:5) |
| Antibody | Anti-CD335-PE (9E2/NKp46) (mouse monoclonal) | BD Biosciences | Cat# 557991, RRID: AB_396974 | FACS (1:20) |
| Antibody | Anti-HLA-DR-FITC (TU36) (mouse monoclonal) | BD Biosciences | Cat# 555560, RRID: AB_395942 | FACS (1:5) |
| Recombinant DNA reagent | pCW57.1 mDux-CA | PMID:28459454 | RRID: Addgene_99284 | All-in-one lentivector with tet-inducible mouse Dux |
| Recombinant DNA reagent | pcDNA 3.1(-) mouse C/EBP delta | Johnson Laboratory | RRID: Addgene_12559 | Mouse C/EBPδ cDNA |
| Recombinant DNA reagent | TRE_3xFlag-C/EBPδ | This paper | N/A | AG Roth (Institute of Immunology, WWU Münster) |
| Recombinant DNA reagent | TRE_ctrl | This paper | N/A | AG Roth (Institute of Immunology, WWU Münster) |
| Recombinant DNA reagent | lentiGuide-puro for GeCKO screen | PMID:25075903 | RRID: Addgene_1000000053 | Lentiviral pooled libraries for genome-wide CRISPR/Cas9 Knock-outs |
| Recombinant DNA reagent | lentiCRISPRv2 | PMID:25075903 | RRID: Addgene_52961 | Lentivector with CRISPR/Cas9 expression |
| Recombinant DNA reagent | lentiCRISPRv2-PHF8 gRNA | This paper | N/A | AG Roth (Institute of Immunology, WWU Münster) |
| Recombinant DNA reagent | lentiCRISPRv2-CSRP1 gRNA | This paper | N/A | AG Roth (Institute of Immunology, WWU Münster) |
| Recombinant DNA reagent | lentiCRISPRv2-HAND1 gRNA | This paper | N/A | AG Roth (Institute of Immunology, WWU Münster) |
| Recombinant DNA reagent | lentiCRISPRv2-FBXW7 gRNA | This paper | N/A | AG Roth (Institute of Immunology, WWU Münster) |
| Recombinant DNA reagent | lentiCRISPRv2-ATF3 gRNA | This paper | N/A | AG Roth (Institute of Immunology, WWU Münster) |
| Recombinant DNA reagent | lentiCRISPRv2-STAT3 gRNA | This paper | N/A | AG Roth (Institute of Immunology, WWU Münster) |
| Recombinant DNA reagent | lentiCRISPRv2-KLF5 gRNA | This paper | N/A | AG Roth (Institute of Immunology, WWU Münster) |
| Recombinant DNA reagent | lentiCRISPRv2-IRF7 gRNA | This paper | N/A | AG Roth (Institute of Immunology, WWU Münster) |
| Recombinant DNA reagent | lentiCRISPRv2-C/EBPβ gRNA | This paper | N/A | AG Roth (Institute of Immunology, WWU Münster) |
| Recombinant DNA reagent | pLenti CMV GFP Blast (659-1) | PMID:19657394 | RRID: Addgene_17445 | Lentiviral eGFP expression vector |
| Recombinant DNA reagent | MSCV-PIG-empty | PMID:29316427 | RRID: Addgene_105594 | Empty control vector with GFP marker and puro resistance |
| Recombinant DNA reagent | MSCV *S100a8* prom_GFP | This paper | N/A | AG Roth (Institute of Immunology, WWU Münster) |
| Recombinant DNA reagent | MSCV *S100a8* s1 prom_GFP | This paper | N/A | AG Roth (Institute of Immunology, WWU Münster) |
| Recombinant DNA reagent | MSCV *S100a8* s2 prom_GFP | This paper | N/A | AG Roth (Institute of Immunology, WWU Münster) |
| Recombinant DNA reagent | MSCV *S100a8* s3 prom_GFP | This paper | N/A | AG Roth (Institute of Immunology, WWU Münster) |
| Recombinant DNA reagent | MSCV *S100a8* s2+3 prom_GFP | This paper | N/A | AG Roth (Institute of Immunology, WWU Münster) |
| Recombinant DNA reagent | MSCV *S100a9* prom_GFP | This paper | N/A | AG Roth (Institute of Immunology, WWU Münster) |
| Recombinant DNA reagent | MSCV *S100a9* s1 prom_GFP | This paper | N/A | AG Roth (Institute of Immunology, WWU Münster) |

*Appendix 1 Continued*

| Reagent type (species) or resource | Designation | Source or reference | Identifiers | Additional information |
|---|---|---|---|---|
| Recombinant DNA reagent | MSCV *S100a9* s2 prom_GFP | This paper | N/A | AG Roth (Institute of Immunology, WWU Münster) |
| Recombinant DNA reagent | MSCV *S100a9* s3 prom_GFP | This paper | N/A | AG Roth (Institute of Immunology, WWU Münster) |
| Recombinant DNA reagent | MSCV *S100a9* s4 prom_GFP | This paper | N/A | AG Roth (Institute of Immunology, WWU Münster) |
| Recombinant DNA reagent | psPAX2 | Trono Laboratory | RRID: Addgene_12260 | Lentiviral packaging plasmid |
| Recombinant DNA reagent | pCMV-VSV-G | Weinberg Laboratory | RRID: Addgene_8454 | Envelope protein for producing lentiviral particles |
| Sequence-based reagent | Cloning primers | This paper | Listed in *Supplementary file 1a, c, d* | |
| Sequence-based reagent | GeCKO library amplification and NGS primers | This paper | Listed in *Supplementary file 1b* | |
| Sequence-based reagent | Mutagenesis primers | This paper | Listed in *Supplementary file 1e* | |
| Sequence-based reagent | qPCR primers | This paper | Listed in *Supplementary file 1f, g* | |
| Peptide, recombinant protein | rm mouse GM-CSF | ImmunoTools | Cat# 12343137 | 40 ng/ml in vitro |
| Peptide, recombinant protein | LPS from *Salmonella enteritidis* | Sigma-Aldrich | Cat# L6011 | 0.5 mg/ml in vivo |
| Peptide, recombinant protein | LPS from *Escherichia coli* O55:B5 | Sigma-Aldrich | Cat# L2880 | 10 ng/ml in vitro |
| Peptide, recombinant protein | rm mouse IL-4 | Peprotech | Cat# 214-14 | 20 ng/ml in vitro |
| Peptide, recombinant protein | rm mouse IFN-γ | ImmunoTools | Cat# 12343536 | 50 ng/ml in vitro |
| Commercial assay or kit | NucleoSpin RNA, Mini kit for RNA purification | Macherey Nagel | Cat# 740955.50 | |
| Commercial assay or kit | QuikChange II XL Site-Directed Mutagenesis Kit | Agilent Technologies | Cat# 200521 | |
| Commercial assay or kit | PureLink HiPure Plasmid Miniprep Kit | Thermo Fisher Scientific | Cat# K210002 | |
| Commercial assay or kit | NEBNext High-Fidelity 2× PCR Master Mix | NEB | Cat# M0541S | |
| Commercial assay or kit | QIAquick PCR Purification Kit | Qiagen | Cat# 28104 | |
| Commercial assay or kit | Bioanalyzer High Sensitivity DNA Analysis Agilent High Sensitivity DNA Kit | Agilent | Cat# 5067-4626 | |
| Commercial assay or kit | 20% PhiX spike in (Illumina PhiX control kit) | Illumina | Cat# FC-110-3001 | |
| Commercial assay or kit | NextSeq 500/550 High Output Kit v2.5 (75 Cycles) | Illumina | Cat# 20024906 | |
| Commercial assay or kit | NEBNext Ultra RNA Library Prep Kit for Illumina | NEB | Cat# E7530 | |
| Commercial assay or kit | LEGENDplex Mouse Inflammation Panel (13-plex) | BioLegend | Cat# 740446 | |
| Chemical compound, drug | Doxycycline | Sigma-Aldrich | Cat# D9891 | 2 µg/ml |
| Chemical compound, drug | Puromycin | InvivoGen | Cat# ant-pr-1 | 3–20 µg/ml |
| Chemical compound, drug | GSK-J4 HCl | SellekChem | Cat# S7070 | 5 µM |
| Chemical compound, drug | β-Estradiol | Sigma-Aldrich | Cat# E8875 | 1 µM |

*Appendix 1 Continued on next page*

*Appendix 1 Continued*

| Reagent type (species) or resource | Designation | Source or reference | Identifiers | Additional information |
|---|---|---|---|---|
| Chemical compound, drug | Polybrene | Sigma-Aldrich | Cat# 107689 | 8 µg/ml |
| Chemical compound, drug | Protease Inhibitor Cocktail | Sigma-Aldrich | Cat# P8340 | 1:1000 |
| Chemical compound, drug | Dynabead Protein G for Immunoprecipitation | Thermo Fisher Scientific | Cat# 10004D | ChIP (1:10) |
| Chemical compound, drug | M-PER Mammalian Protein Extraction Reagent | Thermo Fisher Scientific | Cat# 78501 | 1× |
| Chemical compound, drug | Dihydrorhodamine 123 | Sigma-Aldrich | Cat# D1054 | 15 µm |
| Chemical compound, drug | NEBNext Poly(A) mRNA Magnetic Isolation Module | NEB | Cat# E7490 | |
| Chemical compound, drug | DNA Polymerase I, Large (Klenow) Fragment | NEB | Cat# M0210 | |
| Chemical compound, drug | FluoSpheres Polystyrene Microspheres, 1.0 µm, blue-green fluorescent (430/465) | Thermo Fisher Scientific | Cat# F13080 | $5 \times 10^6$ per $0.5 \times 10^6$ cells |
| Software, algorithm | AliBaba2.1 | PMID:11808873 | http://gene-regulation.com/pub/programs/alibaba2/ | |
| Software, algorithm | FlowJo Software | BD Biosciences | https://www.flowjo.com/solutions/flowjo/ | |
| Software, algorithm | MaGeCK 0.5.9.3 | PMID:25476604 | https://sourceforge.net/p/mageck/wiki/Home/ | |

