## [Editor Report]

The study uses an elegant CRISPR/Cas9 screening approach in a myeloid cell line to identify the underlying regulators of the alarmins S100A8 and S100A9, which amplify inflammation. This approach identified the transcription factor C/EBP-δ as a regulator of S100A8 and S100A9 expression in the myeloid cell line and also showed a correlation between the expression levels of C/EBP-δ and the alarmins in patient samples of peripheral blood mononuclear cells. Furthermore, the authors also validate their findings in primary monocytes using mice with genetic C/EBP-δ deletion. This work will be of significant interest to researchers studying the regulation of immune responses and inflammation, and it also highlights how unbiased CRISPR/Cas9 screening can lead to novel mechanistic insights in myeloid cells.

---

## [Decision Letter]

**Decision letter after peer review:**

Thank you for submitting your article "C/EBPδ-induced epigenetic changes control the dynamic gene transcription of S100A8 and S100A9" for consideration by *eLife*. Your article has been reviewed by 3 peer reviewers, including Jalees Rehman as the Reviewing Editor and Reviewer #1, and the evaluation has been overseen by a Reviewing Editor and Carla Rothlin as the Senior Editor.

Essential revisions:

1) Mechanistic validation of the effects of C/EBP-δ deletion or knockdown in primary human and/or mouse monocytes as outlined in the comments by the reviewers 1 and 2 below. Demonstrating how C/EBP-δ deletion affects the phenotypes and differentiation of primary human or primary mouse monocytes in vivo and in vitro, will validate the conclusions derived from the myeloid cell line.

2) Expanded analysis of the CRISPR/Cas9 screen such as addressing potential false positives (see comments of reviewer 3).

3) Rigorous presentation of the ATAC-seq data including showing all the differential peak analyses and ATAC-seq of the C/EBPδ KO cells (see comments of reviewer 2).

*Reviewer #1 (Recommendations for the authors):*

The manuscript could be strengthened with the following revisions:

1. The authors highlight the importance of alarmins in disease but there are no experiments showing that deletion of C/EBP-δ affects disease processes in vivo. Experiments showing that C/EBP-δ deletion in vivo changes disease and inflammatory injury severity (not just relative circulating levels of alarmins in the serum shown in Figure 2) would be important to highlight the pathophysiological relevance of the C/EBP-δ link to alarmins. The authors can choose an in vivo LPS model as they also use LPS ex vivo and then assess various parameters of in vivo tissue injury and inflammation in C/EBP-δ knockout versus control mice.

2. The phenotyping of the myeloid cells in the absence of C/EBP-δ is very limited and needs to be more rigorous as well as more comprehensive.

a) In Figure 2D, n-fold change in serum levels should be changed to actual alarmin concentrations in the serum but other circulating cytokine levels should also be shown, both at baseline and with stimulation conditions (such as in vivo LPS exposure)

b) How does the C/EBP-δ deletion affect monocyte differentiation and polarization? Flow cytometry of monocyte and macrophage phenotypes with and without C/EBP-δ as well as with and without stimuli that promote monocyte activation, monocyte to macrophage differentiation and macrophage polarization would be more impactful. Ideally, this can be combined with the in vivo studies proposed in comment #1 which addresses C/EBP-δ deletion affecting disease severity because int he same in vivo study, the authors can isolate various cell types and perform flow cytometry characterization.

c) The phagocytosis and ROS assays in Supplement Figure 2 are very interesting because they suggest that alarmin reduction increases phagocytosis capacity. In some ways, this may seem counter-intuitive because it suggests that C/EBP-δ activates inflammatory signaling but suppresses phagocytosis, an important function of immune cells. This is a very important finding and should be moved to the main figures but needs to be studied more robustly showing phagocytosis kinetics and some analysis of how C/EBP-δ would increase phagocytosis. Does C/EBP-δ inhibit the expression of phagocytosis genes? Does C/EBP-δ deletion impact phagocytosis of bacteria (this can be easily assessed using fluorescent bacteria)?

*Reviewer #2 (Recommendations for the authors):*

The data in the cell line show a direct relationship between C/EBPδ and S100A8/A9, however the data in primary cells is only a correlation. It would be important to show, in either human or mouse primary monocytes, that knockdown of C/EBPδ directly affects S100A8/A9.

Regarding the composition of the cell culture, flow cytometry can be performed on each day of differentiation to validate the purity of monocytes and determine whether macrophages or other cells are present, which could affect the analyses.

The ATAC-seq experiments in Figure 6 should be accompanied by a full list of differential peaks between day 0 and day 3 for the reader to browse as a supplementary file. Furthermore, the peaks in day 3 versus day 0 do not seem dramatically different to the eye, which should be quantified. Moreover, the y-axes are not shown for the tracks in 6C. Finally, it would have been interesting to see how C/EBPδ KO monocytes compared to WT in the ATAC-seq data, as well as where day 5 of differentiation would sit in this dataset. From Figure 2A-B, s100a8 and s100a9 expression at day 5 are downregulated in both WT and KO cells. At this time, would the WT and KO cells become more similar to each other, and/or to day 0 cells?

*Reviewer #3 (Recommendations for the authors):*

1) The current study did not identify any previously reported gene regulators. Further work is needed to address this discrepancy. It is well known that CRISPR screens could generate many false negatives, largely due to the design of ineffective gRNAs. The authors should carefully evaluate these discrepant results by experimental validation.

2) I'd like to suggest reanalyzing Figure 1B by focusing on a smaller number (but more significant) of cells. The reduction of screening noise could potentially identify other candidates of interest.

3) I'd like to suggest validating several other candidates to (1) confirm their impact on S100A9 expression, and (2) further demonstrate the robustness of the CRISPR screening system. Although the focus of the current study is on C/EBPD, the inclusion of other candidates would expand the study scope and warrant future studies.

4) The legend for Figure 6A is too small to read and should be enlarged.

---

## [Author Response]

Essential revisions:1) Mechanistic validation of the effects of C/EBP-δ deletion or knockdown in primary human and/or mouse monocytes as outlined in the comments by the reviewers 1 and 2 below. Demonstrating how C/EBP-δ deletion affects the phenotypes and differentiation of primary human or primary mouse monocytes in vivo and in vitro, will validate the conclusions derived from the myeloid cell line.

We now added data obtained from primary cells as requested by the reviewers (Figure 2, D, lines 189 – 190, Figure 2, E – J and Figure 2 —figure supplement 2, lines 190 – 202, Figure 3, F and Figure 3 —figure supplement 2, lines 212 – 224). For details see our point-by-point response to the individual reviewers #1 at points “1 and “2 and #2 at point “2.

2) Expanded analysis of the CRISPR/Cas9 screen such as addressing potential false positives (see comments of reviewer 3).

We addressed this comment in our revised manuscript (see Figure 1 —figure supplement 1, A, lines 153 – 161 in the manuscript and at points “1, “2 and “3 in response to reviewer #3).

3) Rigorous presentation of the ATAC-seq data including showing all the differential peak analyses and ATAC-seq of the C/EBPδ KO cells (see comments of reviewer 2).

The primary goal of our study was not the effect of C/EBPδ on the genome wide genome structure but its role in the expression of *S100a8* and *S100a9*. However, we now performed and added an additional genome-wide ATAC-seq analysis of C/EBPδ KO monocytes and present all data in the Figure 7, A – C, Figure 7 —figure supplement 1 and Figure 7 – source data 1 as supplemental material as requested by reviewer #2 (see lines 278 – 287 in the manuscript and our point “3 in response to reviewer #2).

Reviewer #1 (Recommendations for the authors):The manuscript could be strengthened with the following revisions:1. The authors highlight the importance of alarmins in disease but there are no experiments showing that deletion of C/EBP-δ affects disease processes in vivo. Experiments showing that C/EBP-δ deletion in vivo changes disease and inflammatory injury severity (not just relative circulating levels of alarmins in the serum shown in Figure 2) would be important to highlight the pathophysiological relevance of the C/EBP-δ link to alarmins. The authors can choose an in vivo LPS model as they also use LPS ex vivo and then assess various parameters of in vivo tissue injury and inflammation in C/EBP-δ knockout versus control mice.

We agree with this objection. Therefore, we analysed the role of the C/EBPδ-S100A8/A9 axis in a mouse model for acute lung inflammation, which is characterized by high S100A8/A9-level. We could show that LPS-exposed C/EBPδ deficient mice secrete less S100A8/A9 systemically (serum) and locally (bronchoalveolar lavage fluid, BALF) (Figure 2, E and F). Lower S100A8/A9-levels are accompanied by lower disease activity reflected by decreased local cytokine production (Figure 2, H – J), highlighting the influence of C/EBPδ-deficiency on alarmin expression and inflammatory injury in vivo.

2. The phenotyping of the myeloid cells in the absence of C/EBP-δ is very limited and needs to be more rigorous as well as more comprehensive.

To go more in depth on the phenotype of C/EBPδ-deficient myeloid cells, we examined monocyte-to-macrophage differentiation of WT and C/EBPδ KO bone marrow-derived cells and ER-Hobx8 cells using flow cytometric analysis (Figure 3, F). Analysis of the proportion of CD11b^+^Ly-6C^+^ cells reveals no differences between WT and C/EBPδ KO cells during differentiation, which is a prerequisite for C/EBPδ KO cells to pass the pre-gating on CD11b^+^Ly-6C^+^ within the GeCKO library screen to exclude phenotypes that are S100A9^low/neg^ due to differentiation defects. In addition, we compared differentiation to M1 and M2 macrophages between wild type and C/EBPδ KO mice (Figure 3 —figure supplement 2).

a) In Figure 2D, n-fold change in serum levels should be changed to actual alarmin concentrations in the serum but other circulating cytokine levels should also be shown, both at baseline and with stimulation conditions (such as in vivo LPS exposure)

To examine alarmin secretion at baseline (NaCl) and after LPS-exposure we now added new data obtained from a mouse model of acute lung inflammation. We analysed serum and bronchoalveolar lavage fluid (BALF) for S100A8/A9 (Figure 2, E and F) using our in-house ELISA as described. In addition, we analysed inflammatory cytokines, such as IL-1α, IL-6 and TNF-α in BALF (Figure 2, H – J), and various further cytokines (Figure 2 —figure supplement 2) in WT and C/EBPδ KO sera at both conditions, using a bead based immunoassay (LEGENDplexTM mouse inflammation panel, BioLegend).

b) How does the C/EBP-δ deletion affect monocyte differentiation and polarization? Flow cytometry of monocyte and macrophage phenotypes with and without C/EBP-δ as well as with and without stimuli that promote monocyte activation, monocyte to macrophage differentiation and macrophage polarization would be more impactful. Ideally, this can be combined with the in vivo studies proposed in comment #1 which addresses C/EBP-δ deletion affecting disease severity because int he same in vivo study, the authors can isolate various cell types and perform flow cytometry characterization.

We analysed whether C/EBPδ deletion affects polarization of bone-marrow derived monocytes stimulated either with IFN-γ + LPS (M_1_) or IL-4 (M_2_) for 24 hours. We found that expression of M_1_-monocyte associated markers, such as *Tnfa*, *Il6*, *Inos*, *Cd86* and *Il1b*, was only slightly decreased in LPS and IFN-γ treated C/EBPδ KO monocytes (Figure 3 —figure supplement 2, A), whereas expression of *Il10* was diminished, but M_2_-associated *Cd163* expression showed no significant reduction after IL-4-stimulation in C/EBPδ KO cells (Figure 3 —figure supplement 2, B). As mentioned above, the new data set of genome-wide ATAC-seq maybe a useful source for analysis of C/EBPδ effects on chromatin remodelling during macrophage differentiation (Figure 7, A – C, Figure 7 —figure supplement 1 and Figure 7 – source data 1 as supplemental material).

c) The phagocytosis and ROS assays in Supplement Figure 2 are very interesting because they suggest that alarmin reduction increases phagocytosis capacity. In some ways, this may seem counter-intuitive because it suggests that C/EBP-δ activates inflammatory signaling but suppresses phagocytosis, an important function of immune cells. This is a very important finding and should be moved to the main figures but needs to be studied more robustly showing phagocytosis kinetics and some analysis of how C/EBP-δ would increase phagocytosis. Does C/EBP-δ inhibit the expression of phagocytosis genes? Does C/EBP-δ deletion impact phagocytosis of bacteria (this can be easily assessed using fluorescent bacteria)?

We thank the reviewer for pointing to the importance of the effect of C/EBPδ-deficiency on phagocytosis-rates. We added data on phagocytosis of bacteria and found higher phagocytosis capacities of fluorescent *Staphylococcus aureus* (Figure 3, C and D) as well. Analysis of Pattern Recognition Receptors-gene expression revealed that *Ptx3*, *Dc-sign* or *Cd14* are up-regulated in C/EBPδ KO cells, which may explain higher phagocytosis rates due to increased pathogen-recognition (Figure 3 —figure supplement 1). C/EBPδ could act as a suppressor/regulator of phagocytosis, but these mechanisms need further investigation which is behind the scope of our study on regulatory mechanisms of *S100* expression. For our question, the most important observation was that C/EBPδ-deficiency does not dampen inflammatory responses in general but that its impact on S100A8 and S100A9 expression is rather specific. However, we agree with the comment of the reviewer that the phagocytosis and ROS assays reveal very interesting facts, which is why we moved these findings to the main manuscript as suggested.

Reviewer #2 (Recommendations for the authors):The data in the cell line show a direct relationship between C/EBPδ and S100A8/A9, however the data in primary cells is only a correlation. It would be important to show, in either human or mouse primary monocytes, that knockdown of C/EBPδ directly affects S100A8/A9.

Please see our response to point “2.

Regarding the composition of the cell culture, flow cytometry can be performed on each day of differentiation to validate the purity of monocytes and determine whether macrophages or other cells are present, which could affect the analyses.

We agree that validation of monocyte purity is very important at this point, because altered differentiation kinetics would influence S100A8 and S100A9 expression. Flow cytometric analysis of WT and C/EBPδ KO cells reveals neither differences in the purity of CD11b^+^Ly-6C^+^ cells nor of differentiation kinetics (Figure 3, F), which is a prerequisite to compare time-point dependent S100A8 and S100A9 level of WT and C/EBPδ KO cells as we did.

The ATAC-seq experiments in Figure 6 should be accompanied by a full list of differential peaks between day 0 and day 3 for the reader to browse as a supplementary file. Furthermore, the peaks in day 3 versus day 0 do not seem dramatically different to the eye, which should be quantified. Moreover, the y-axes are not shown for the tracks in 6C. Finally, it would have been interesting to see how C/EBPδ KO monocytes compared to WT in the ATAC-seq data, as well as where day 5 of differentiation would sit in this dataset. From Figure 2A-B, s100a8 and s100a9 expression at day 5 are downregulated in both WT and KO cells. At this time, would the WT and KO cells become more similar to each other, and/or to day 0 cells?

We are thankful and agree for this comment. Indeed, we completed our ATAC-seq data by adding C/EBPδ KO cells on day 0 and day 3 (n = 3) to the analysis and a comprehensive list of all regions with differential peaks between all four conditions (WT and C/EBPδ KO cells on day 0 and day 3) (Figure 7, A – C, Figure 7 —figure supplement 1, Figure 7 – source data 1). In our revised manuscript, we also now statistically analysed differential chromatin accessibility between all conditions by specifying the adjusted p-values (padj) < 0.05 and log2 fold changes (Figure 7, C). We agree with the comment of the reviewer raised above that adding data of C/EBPδ KO cells fills a gap between ATAC-seq and ChIP data.

Reviewer #3 (Recommendations for the authors):1) The current study did not identify any previously reported gene regulators. Further work is needed to address this discrepancy. It is well known that CRISPR screens could generate many false negatives, largely due to the design of ineffective gRNAs. The authors should carefully evaluate these discrepant results by experimental validation.

This point is a very important and justifiable note. The impact of previously reported gene regulators, such as ATF3, STAT3, KLF5, IRF7 or C/EBPβ on *S100a8* and *S100a9* expression was examined in our study by creating specific KO cell lines of these candidates. None of these suggested genes caused a relevant reduction of *S100a8* and *S100a9* expression upon deletion (former: Figure 2 —figure supplement 3, now: Figure 1 —figure supplement 1, B), thus confirming their absence in the CRISPR screen. In addition, our screening approach intentionally excluded factors which induce a general block of monocyte differentiation to exclude indirect effects not related to direct transcriptional regulation of these two S100 genes. However, transcription factors which show both, effects on general differentiation and *S100* expression, will not be detected in our screen. We discuss this point in our manuscript (lines 141 – 142, 349 – 352)*.* Taken together, in our opinion CRISPR screens have major limitations which do not allow reliable identification of a network of factors regulating gene expression but can be very useful for unbiased screening for individual factors. The latter has to be confirmed by independent validation experiments as done here for C/EBPδ.

2) I'd like to suggest reanalyzing Figure 1B by focusing on a smaller number (but more significant) of cells. The reduction of screening noise could potentially identify other candidates of interest.

We agree that one might retrospectively argue that a different gating strategy may have improved our screening approach, but reanalysis of the existing data is not feasible since cells were already sorted, processed and analysed in the CRISPR screen. However, analysis of our actual sorted cell populations (hits negative and reference positive) by immunoblotting for S100A9 (Figure 1, B) clearly validates our gating strategy and excludes gating as a major bias of our results.

3) I'd like to suggest validating several other candidates to (1) confirm their impact on S100A9 expression, and (2) further demonstrate the robustness of the CRISPR screening system. Although the focus of the current study is on C/EBPD, the inclusion of other candidates would expand the study scope and warrant future studies.

We agree with this point and present now additional data showing that single knock-out cell lines for further hits as PHF8, CSRP1 or HAND1 do not dampen *S100a8* and *S100a9* expression extensively, especially in comparison to C/EBPδ KO cells (Figure 1 —figure supplement 1, A, lines 159 – 161).

4) The legend for Figure 6A is too small to read and should be enlarged.

Thank you for this annotation.